# The Positive Effects of Employee AI Dependence on Voice Behavior—Based on Power Dependence Theory

**DOI:** 10.3390/bs15121709

**Published:** 2025-12-10

**Authors:** Jialin Liu, Mingpeng Huang, Min Cui, Guangdi Tian, Xinyue Li

**Affiliations:** 1Business School, University of International Business and Economics, Beijing 100029, China; 202200320077@uibe.edu.cn (J.L.); hmp@uibe.edu.cn (M.H.); 2School of Management, Northwest Normal University, Lanzhou 730070, China; 3School of Business Administration, Shanxi University of Finance and Economics, Taiyuan 030006, China; tgd@sxufe.edu.cn; 4Party School of State Grid Corporation of China (Senior Management Training Center of State Grid Corporation of China), Beijing 100192, China; longlixinyue@163.com

**Keywords:** artificial intelligence, power dependence theory, employee voice behavior, coaching leadership, AI-era leadership

## Abstract

The rapid integration of artificial intelligence (AI) into organizational workflows is re-shaping traditional patterns of interaction between leaders and employees. Grounded in power dependence theory, this study investigates how employees’ voluntary dependence on AI influences leader–subordinate power relations and, consequently, influences employees’ voice behavior. We propose that employees’ dependence on AI can increase their perceived power when interacting with leaders, which subsequently enhances their willingness to offer constructive suggestions or question established practices. Furthermore, we propose that the extent to which leadership tasks can be substituted by AI plays a moderating role in this process. Coaching leadership, characterized by its emphasis on guiding task performance and developing employee skills, may be particularly sensitive to such substitution. Using two experimental studies and two survey investigations, we provide evidence that employees’ AI dependence is positively associated with voice behavior through heightened perceptions of personal power, and that this relationship is strengthened under high levels of coaching leadership. These findings advance leadership theory by explicating how AI adoption alters foundational power structures in the workplace and by identifying a novel, power-based pathway linking AI use to proactive employee behaviors. The study contributes to emerging discussions on effective leadership in technologically augmented organizations and offers empirical insights into how leaders can adapt their roles and behaviors in the new era of AI-driven work.

## 1. Introduction

The contemporary workplace is experiencing significant changes as Artificial Intelligence (AI) becomes increasingly embedded in organizational activities. AI systems now autonomously perform a wide range of cognitive and analytical tasks ([88]). These technologies improve the quality of decision-making, simplify operational routines, and generate actionable insights across various sectors ([13]; [14]). As intelligent tools are incorporated into job design, employees have begun to depend more on AI in areas such as task execution, decision support, and performance monitoring ([74]; [89]).

Existing studies have primarily focused on the outcomes of AI usage, including its effects on performance, autonomy, innovation, and job security ([2]; [87]; [75]; [93]; [58]). AI usage typically refers to the frequency and intensity with which employees interact with AI systems in their daily work ([87]). However, as AI becomes more deeply embedded within organizational processes, employees may develop a deeper psychological dependence on these technologies. In practice, this is evidenced by a growing tendency among employees to delegate core work components—such as composing emails and reports, analyzing data, generating strategic ideas, and even drafting performance reviews—to generative AI tools, indicating a phenomenon called “AI dependence syndrome” ([21]). Moreover, in workplaces, AI has exhibited the capability to take on leadership roles ([35]). For example, an AI system named “Mika” was appointed interim CEO at Dictador in 2022 ([61]). More recently, this trend reached a new milestone when Albania appointed an AI named “Diella” as the Minister of Public Procurement in 2025, formally vesting it with governmental power to evaluate bids and combat corruption ([34]). Despite such emblematic developments, limited attention has been given to how dependence on AI influences interpersonal relationships between employees and their human supervisors.

Following [88] ([88]), AI dependence is defined as employees’ perception that effectively accomplishing their work requires reliance on the advanced capabilities of AI systems. Unlike AI usage, which emphasizes how often AI is used, dependence emphasizes how essential and irreplaceable AI is perceived to be in enabling decision-making and task success. This conceptualization treats AI dependence as a state-like and situational construct that may vary across work contexts and over time, consistent with prior views that regard dependence as episodic and context-specific ([80]; [81]). Accordingly, AI dependence captures a deeper level of human–technology interaction, offering new insight into how employees’ psychological perceptions and social relationships may shift as dependence on intelligent systems increases. Despite its growing importance, research on AI dependence remains limited, particularly concerning its implications for relationships between leaders and employees. This study aims to fill this gap by examining how employees’ dependence on AI reshapes the power structure and interaction patterns within leader–subordinate relationships. Understanding these changes can assist organizations in adjusting leadership practices to accommodate AI’s expanding presence in daily work.

Grounded in Power Dependence Theory ([26]), this study explores how the integration of AI influences power relations between employees and leaders. The theory suggests that one party’s (X) influence over another (Y) depends on Y’s reliance on resources controlled by X and the availability of alternatives. Traditionally, employees depend on leaders for task-related resources ([40]). As AI now provides these forms of support, including information processing, decision assistance, and performance verification ([88]), it enhances employees’ access to resources and feedback, partly substituting leadership functions ([46]). This shift may reduce employees’ dependence on leaders, thereby altering perceptions of leader power and enhancing employees’ sense of personal power in their interactions with supervisors ([26]). Such changes in perceived power may subsequently shape employee behavior in the workplace ([64]).

According to Power-Dependence Theory ([26]), power arises from the dependency in social exchange relationships. When employees gain more resources through AI—such as decision support, feedback, and skill enhancement—and reduce their dependence on leaders, their sense of power in interactions with leaders increases. Changes in power-dependence relationships often first appear in interaction patterns ([63]), which may then influence individuals’ social behaviors and willingness to exert influence ([6]). Voice behavior refers to employees’ voluntary efforts to offer ideas or suggestions to improve organizational operations ([67]; [73]; [92]). This behavior is both constructive and socially risky, as it reflects employees’ organizational commitment and motivation for improvement, but it can also be seen as a challenge to existing authority and decision-making ([66]). In contrast, task-oriented behaviors, such as creativity or performance, reflect an individual’s use of resources and ability performance, rather than the redistribution of power within social relationships ([42]; [10]). Therefore, voice behavior is more capable of revealing the relational changes triggered by AI dependence, which reflects employees’ repositioning and proactive influence tendencies in leader-employee relationships as their sense of power increases ([90]). As employees feel more empowered, they are more willing to take social risks and proactively voice their opinions, thus promoting organizational improvement ([73]; [16]; [67]; [56]). By providing reliable information and feedback, AI partially substitutes for leadership functions, alter the patterns of employees’ dependence on leaders, and increase employees’ autonomy and informational control in leader interactions, thereby enhancing their perceptions of power and encouraging more frequent voice behavior.

The substitutability of resource channels is crucial in understanding how power shifts occur ([26]). Power-Dependence Theory suggests that the power imbalance between two parties is determined by their control over and access to valued resources. As AI tools increasingly perform tasks that were traditionally carried out by leaders, such as providing feedback, supporting decision processes, and facilitating employee development, employees may become less reliant on their leaders, contributing to changes in power relations ([70]; [94]) Coaching leadership is especially relevant in this context because it focuses on helping employees improve skills, receive guidance, and develop competence ([5]; [49]; [44]; [100]). These functions overlap considerably with the developmental support that AI systems can provide ([79]). When coaching leadership is high, the functional overlap between leadership and AI is more pronounced, thereby amplifying the impact of AI dependence on employees’ perceived power. In contrast, when coaching leadership is low, AI is less able to substitute for leadership functions, and the effect of AI dependence on perceived power is likely to be diminished. Thus, we propose that the degree of coaching leadership moderates the relationship between AI dependence and employees’ perceived power, with the relationship is stronger under high levels of coaching leadership than under low levels.

This study makes a significant theoretical contribution by examining how employees’ increasing dependence on AI reshapes power dynamics within leader-employee relationships, drawing on power dependence theory. By introducing AI as a third-party actor, it highlights how power can be redistributed among leaders, employees, and AI systems. The findings indicate that employees’ growing dependence on AI enhances their sense of personal power in interactions with leaders, thereby altering traditional power dynamics and interaction patterns. Furthermore, the study reveals that coaching leadership moderates this relationship, intensifying the impact of AI dependence on employees’ perceived power. This finding extends existing leadership theories by suggesting that AI can substitute certain interpersonal leadership functions. The study also offers practical insights for organizations to adapt leadership practices in the AI era while maintaining effective leader-employee relationships.

## 2. Theory and Hypotheses

### 2.1. Power Dependence Theory

In organizational contexts, power dynamics often shape the relationship between employees and leaders. Power is defined as the ability to influence goal achievement and to control access to valued resources ([45]). Leaders traditionally possess greater power because their higher hierarchical position allows them to allocate resources such as decision authority, performance feedback, and task assignments ([48]; [55]). Employees, on the other hand, have historically relied on leaders for these resources to accomplish their work-related goals ([96]). According to Power Dependence Theory ([26]), power in such relationships is determined by the extent to which one party relies on another for critical resources, resulting in an inherent asymmetry when dependence is high.

However, the introduction of AI into the workplace has shifted these traditional power structures ([65]). AI systems, such as intelligent assistants, are increasingly performing functions that were once the exclusive domain of leaders, including decision support, feedback provision, and performance monitoring ([60]). This allows employees to access alternative resources, reducing their reliance on leaders and thus altering the traditional power structure. As AI takes on more leadership functions, employees may experience greater autonomy and control over their work processes, which diminishes their dependence on leaders for guidance ([70]; [94]). These developments underscore the importance of examining how AI dependence reshapes leader–employee interactions. By applying Power Dependence Theory, we can better understand how these changes in power dependencies influence employee behaviors, particularly in terms of their interactions with leaders.

### 2.2. Employee Dependence on AI and Personal Sense of Power

The integration of AI is reshaping power-dependence relationships within organizations. According to Power-Dependence Theory, power within an exchange relationship is determined by the extent to which one party depends on another for valued resources ([26]). Power, therefore, is not a fixed personal attribute but a feature of the relationship itself, shaped by the pattern of mutual dependence between interacting parties. In organizational settings, this dependence is often vertical, as employees rely on leaders for access to information, resources, decision authority, and performance feedback ([96]; [85]).

When alternative sources of resources become available, the dependence of the previously subordinate party decreases, and their relative power increases ([26]). In this sense, AI increasingly functions as an alternative source of leadership resources by performing managerial, informational, and cognitive tasks such as scheduling, task delegation, feedback provision, and performance monitoring ([51]; [97]; [91]). By supporting employees in managing and executing work in ways comparable to human leaders ([31]), AI reduces their reliance on supervisors for direction, expertise, and feedback. This shift alters the asymmetrical power structure that traditionally characterizes leader–follower interactions. This substitution is particularly evident in domains requiring information and expertise. AI provides accurate, objective, and comprehensive recommendations ([14]; [41]), mitigating the informational asymmetry that historically reinforces leaders’ authority ([59]). With access to clearer and more transparent information provided by AI systems, employees gain greater control over task-relevant knowledge and decision-making resources. Consequently, as they acquire these alternative informational channels, they experience enhanced autonomy and occupy a stronger position in interpersonal exchanges with their leaders ([4]; [83]).

Beyond informational substitution, AI enhances employees’ autonomy and efficacy in executing tasks. By supporting complex problem-solving and offering instantaneous, objective feedback ([74]; [89]; [97]), AI reduces the uncertainty and subjective bias often associated with human supervision ([35]). As a result, employees become more self-sufficient; they are able to make decisions, allocate resources, and evaluate task outcomes without constant involvement from their leaders. Consistent with Power-Dependence Theory, as employees gain alternative means to achieve their goals, their relative power in the leader–employee relationship increases. This rise in perceived power stems not simply from a decline in leader authority, but from employees’ expanded control over valued resources and their stronger sense of autonomy and competence ([45]; [6]; [85]). Hence, AI dependence reflects not passive reliance but an active empowerment process through which employees access alternative resources that were once primarily governed by leaders, thereby strengthening their personal sense of power in interactions with leaders.

As AI provides reliable task accomplishment and informational support, it reshapes the power dynamics between leaders and employees by increasing employees’ sense of personal power. We therefore hypothesize:

**H1.** 
*Employee’s dependence on AI is positively correlated with their personal sense of power in interactions with leaders.*


### 2.3. The Mediation Effect of Personal Sense of Power in Interactions with Their Leader

According to Power Dependence Theory ([26]), power emerges from the asymmetrical dependence within social exchange relationships. When individuals gain alternative access to valuable resources, their dependence on others decreases, thereby reducing relational constraints and expanding their behavioral freedom. Individuals with relatively more power are therefore less subject to external control and more able to pursue their own judgments and goals ([64]). These behavioral changes stem from shifts in the underlying structure of dependence.

Building on this foundation, personal sense of power reflects an individual’s internal perception of their influence within the relationship ([45]; [7]). As [7] ([7]) argue, such beliefs can shape their actual influence beyond what is dictated by formal roles or structural positions. Those who perceive themselves as powerful tend to exhibit more confidence and proactive behaviors, which in turn activate their approach systems and encourage them to take initiative and pursue goals with greater autonomy ([8]; [15]). Accordingly, employees who experience a stronger sense of power are more likely to view themselves as the power-holder with access to the opportunities needed to accomplish their objectives ([54]).

In this context, employees’ voice behavior is also significantly influenced by their personal sense of power. Voice behavior refers to employees’ discretionary upward communication aimed at improving work processes ([92]; [20]). Because voice often challenges existing norms and involves interpersonal risk, employees with lower perceived power may remain silent due to concerns about negative evaluation or possible sanctions ([68]; [95]). In contrast, employees who perceive themselves as having higher power (due to reduced dependence on leaders through AI) are more confident in expressing their thoughts and initiating constructive changes. As noted, by [52] ([52]), employees who believe they hold more power infer that they have adequate resources and opportunities to be heard effectively. Consequently, employees with a stronger sense of power are more likely to speak up in the workplace.

Based on this reasoning, we propose that employees’ perceived power in leader interactions positively correlates with voice behavior. Thus, extending H1, we advance the following hypothesis:

**H2.** 
*Employee’s dependence on AI is positively correlated with employee voice behavior through the individual’s personal sense of power in interactions with their leader.*


### 2.4. The Moderate Effect of Coaching Leadership

According to Power-Dependence Theory ([26]), power asymmetry between two parties depends on the control and substitutability of valued resources. When alternative sources become available, dependence on the original resource holder decreases, leading to a redistribution of power. In organizational settings, leaders serve as a key source of valued resources, such as developmental guidance, performance feedback, and informational support, where employees rely on to accomplish their goals. As AI begins to replicate or substitute these leadership functions, employees’ reliance on leaders decreases. This reduced dependence enhances employees’ perceived power when interacting with their leaders.

Among different leadership styles, coaching leadership is particularly relevant in this dynamic because of its strong focus on employee development and empowerment. Coaching leadership emphasizes supporting employees through open communication, guidance, and encouragement for growth ([22]; [57]). It offers essential resources and informational support that enable employees to better understand organizational processes and increase their sense of control over job tasks, thereby boosting engagement and motivation ([18]). Through problem-solving assistance, constructive feedback, and the solicitation of employee input, coaching leaders promote a mutually beneficial exchange that enhances employees’ competence and sustained energy at work ([25]; [24]; [100]). Moreover, coaching leadership fosters alignment between personal and organizational goals and clarifies role expectations ([9]), contributing to a developmental climate characterized by trust and autonomy.

However, the same functions emphasized in coaching leadership, such as developmental feedback, informational guidance, and empowerment, are also areas in which AI technologies have become increasingly capable. Contemporary AI systems can analyze performance data, offer real-time feedback, and generate customized suggestions for skill development ([14]; [30]). When leaders display a high level of coaching leadership, employees are more likely to perceive a functional overlap between AI and their supervisors. This overlap enhances the substitutability of AI as an alternative resource channel, further reducing employees’ dependence on leaders and increasing their perceived personal power in leader–employee interactions. Conversely, when coaching leadership is low, the overlap between AI’s capabilities and leadership behaviors is limited, and AI is therefore less likely to influence power relations.

We further propose that coaching leadership moderates not only the direct relationship between AI dependence and employees’ perceived power but also the indirect effect on voice behavior through personal sense of power. Under conditions of high coaching leadership, AI dependence is more strongly associated with increases in perceived personal power, which subsequently encourages employees to engage in proactive, upward voice behavior. Based on these considerations, we advance the following hypothesis:

**H3.** 
*Coaching leadership moderates the relationship between employees’ dependence on AI and their personal sense of power, indicating that this relationship is stronger when coaching leadership is high rather than low.*


In conjunction with Hypothesis 2, we additionally propose:

**H4.** 
*Coaching leadership moderates the indirect effect of employees’ dependence on AI on voice behavior through personal sense of power, indicating that the indirect effect is stronger when coaching leadership is high compared with low levels.*


In summary, the research model is visually represented in Figure 1.

## 3. Overview of Studies

To provide a comprehensive and rigorous test of our theoretical model, we conducted four complementary studies across different methods, samples, and cultural contexts. This multi-study approach was designed to strengthen both internal and external validity while minimizing the risk of methodological bias. Importantly, because the meanings of power and hierarchical dependence may vary across cultural contexts, particularly in high power-distance societies such as China ([36]; [27]). Testing our model in both the United States and China allows us to assess whether the effects of AI dependence on power dynamics are universal or culturally contingent. Collectively, the studies create a progressive validation framework that combines experimental control with field realism and cross-cultural comparison. Study 1 employed an experimental design with a U.S. employee sample to establish the causal effect of AI dependence on employees’ perceived power (H1). Study 2, conducted in China, replicated this design and further incorporated the mediating and behavioral components of our model (H2). Although voice behavior was measured by self-report, this study extended the mechanism identified in Study 1 and provided initial evidence for mediation within a controlled setting. Study 3 adopted a two-wave survey among full-time employees in the United States to enhance external validity and demonstrate that the observed effects persist beyond experimental manipulation. Finally, Study 4, a field study conducted in China, further validated our theoretical model in a natural organizational context, offering additional support for its practical relevance and cross-cultural applicability. Taken together, these studies integrate experimental control with field realism, showing that the effects of AI dependence are consistent across research designs, measures, and cultural contexts.

## 4. Study 1 Method

### 4.1. Participants and Procedures

In Study 1, participants were recruited from the United States through Prolific, a platform widely used and validated in academic research ([76]; [82]). A priori power analysis indicated that at least 128 participants were required. ([29]). To ensure data quality and reduce the likelihood of inattentive or incomplete responses, we recruited 200 participants, each compensated with £0.40 (approx. $0.51 USD at the time). Following the recommendations of [62]’s ([62]), we embedded an attention check item to monitor participant attentiveness. Participants who failed to pass the attention check or submitted incomplete responses were excluded from the final analysis. Twenty-eight participants were removed, resulting in a final sample of 172 individuals. The sample comprised 82 females (47.7%), with a mean age of 42.94 years (SD = 11.89). In terms of ethnicity, 69.8% identified as Caucasian, 18% as African American, 6.4% as Asian, 1.2% as Hispanic American, and 2.3% did not specify. Participants reported an average of 16.37 years of education (SD = 2.86) and an average organizational tenure of 9.79 years (SD = 8.77).

Participants were randomly assigned to one of two conditions using Prolific’s randomization feature. In the AI Dependence group (n = 87), task completion required the use of an intelligent algorithm ([12]). In the Control condition (n = 85), participants completed the task independently. After the initial survey, each participant was presented with a scenario in which they acted as a consultant at a management firm. Adapted from [88] ([88]), the lemonade stand business simulation was used as the experimental task. In the AI Dependence condition, recommendations were based on AI-provided data updates regarding lemonade characteristics, whereas the Control group operated without assistance. All participants completed the consulting task under their assigned condition.

### 4.2. Manipulations

AI dependence. Following the experimental framework developed by [88] ([88]), participants were asked to imagine themselves as consultants in a management consulting firm and advising a client on strategic decisions for a new lemonade business.

In the AI dependence condition (n = 87), participants collaborated with an intelligent algorithm through a series of visualized conversational interfaces embedded in the survey. The algorithm provided data-driven recommendations and feedback, such as highlighting how variations in sugar, lemon, and color levels would affect customer satisfaction and profitability. In certain instances, it also flagged when a participant’s choice was suboptimal based on its analytical assessment. These interactions required participants to depend on the AI’s informational input to make informed recommendations.

In the control condition (n = 85), participants completed the same decision-making task independently, making all recommendations based on their own reasoning and knowledge without algorithmic assistance. Aside from the presence or absence of AI assistance, the task structure, decision sequence, and available information were identical across the two conditions.

To further enhance immersion and simulate a realistic leader–employee interaction, participants were then asked to respond to an open-ended reflection question: “Please explain the logic behind your consulting recommendation and use it as supporting material to convince your supervisor to adopt your proposal (at least 50 words).” This task encouraged participants to elaborate on their reasoning as if presenting their ideas to a leader, helping them mentally simulate an authentic interaction with a supervisor rather than relying on abstract assumptions or prior personal experiences.

### 4.3. Measures

Employee’s personal sense of power toward the leader was assessed using the eight-item scale developed by [7] ([7]). Participants responded on a five-point Likert scale (1 = strongly disagree, 5 = strongly agree). A sample item is: “In my interactions with my supervisor, I can get him/her/them to listen to what I say.” The scale demonstrated high reliability (Cronbach’s α = 0.89).

AI dependence manipulation check was conducted using a three-item scale adapted from [88] ([88]). Participants indicated their agreement on a five-point scale (1 = strongly disagree, 5 = strongly agree). An example item is: “I depended on AI to manage or assist with work-related tasks.” The scale showed excellent reliability (α = 0.96).

## 5. Study 1 Results

### 5.1. Manipulation Check

The manipulation check confirmed that our experimental manipulation was effective. Participants in the AI Dependence Group reported a significantly higher level of AI dependence (M = 3.88, SD = 0.91) than those in the control condition (M = 1.81, SD = 1.02; *t*_[170]_ = 14.10, *p* < 0.001).

### 5.2. Tests of Hypotheses

Descriptive statistics are presented in Table 1. Hypothesis 1 proposed that there is a positive correlation between employee dependence on AI and their sense of power in interactions with leaders. The ANOVA results indicated that individuals in the AI dependency group reported a significantly higher sense of personal power (*M* = 3.49, *SD* = 0.62) compared to those in the control group (*M* = 3.28, *SD* = 0.69; F _[1, 170]_ = 4.50, *p* < 0.05). Consequently, the results support the Hypothesis 1.

In conclusion, Study 1 provides empirical evidence supporting a causal link between employees’ dependence on AI and their personal sense of power towards their leaders. However, Study 1 examined only a segment of our theoretical framework and relied solely on a U.S. sample. To address these limitations, Study 2 is designed to investigate the mediating effects within our model.

## 6. Study 2 Method

### 6.1. Participants and Procedures

In Study 2, participants were recruited via Credamo, a Chinese online research platform widely used in academic studies ([53]; [99]). The sample consisted of full-time employees. A priori power analysis indicated at least 128 participants were required ([29]). To minimize issues related to inattentive or incomplete responses, we recruited 200 participants, each compensated 2 RMB (approximately $0.30 USD). An attention check, consistent with the criteria used in Study 1, was implemented to ensure data quality. Participants who failed the attention check were excluded. After excluding 2 participants, a total of 167 fully completed questionnaires were collected, resulting in a final sample of 165 respondents (64.8% female). On average, participants had 15.93 years of education (SD = 2.03), 7.19 years of organizational tenure (SD = 6.87), 3.88 years of experience working with their current leader (SD = 3.56), and 2.43 years of AI usage experience (SD = 2.15). As in Study 1, participants were randomly assigned to one of two conditions: the AI Dependence group (n = 82), in which involved collaboration with AI ([12]), and the Control group (n = 83), in which participants worked independently without AI support. After being recruited via the Chinese research platform Credamo, participants were randomly assigned to either group using the platform’s built-in randomization function.

### 6.2. Manipulations

AI dependence. The manipulation procedure for AI dependence remained the same as that used in Study 1.

### 6.3. Measures

Employee’s personal sense of power toward the leader was assessed using the same eight-item scale as in Study 1 ([7]). Cronbach’s α for this scale was 0.87.

Employee voice behavior was measured with a three-item scale developed by [50] ([50]). Before responding, participants were reminded: “Based on the scenario you just completed, please rate how likely you would be to engage in the following behaviors.” A sample item is: “How frequently do you voice suggestions for new work-related policies and procedures?” Responses were recorded on a five-point frequency scale (1 = not at all, 5 = extremely). The scale showed acceptable reliability (Cronbach’s α = 0.65).

AI dependence manipulation check was conducted using the same three-item scale as in Study 1 ([88]), which demonstrated good reliability (Cronbach’s α = 0.90).

## 7. Study 2 Results

### 7.1. Manipulations Check

Participants in the experimental group designed to increase AI dependence reported a significantly higher degree of AI dependence (*M* = 4.11, *SD* = 0.54) than those in the control condition (*M* = 1.99, *SD* = 0.75, *t*_[163]_ = 20.83, *p* < 0.001). This indicates that our manipulation was effective.

### 7.2. Tests of Hypotheses

Descriptive statistics and reliability indices are presented in Table 2. In line with hypothesis, ANOVA results indicated that participants in the AI dependence group reported a significantly higher sense of power toward their leaders (*M* = 3.78, *SD* = 0.68) than those in the control group (*M* = 3.54, *SD* = 0.56), *F*(1, 163) = 3.14, *p* < 0.001. Hypothesis 2 proposes that personal sense of power would mediate the relationship between AI dependence and voice behavior. To test this hypothesis, a bootstrap analysis with 5000 resamples was conducted. The results revealed a significant indirect effect (*b* = 0.04, 95% CI [0.003, 0.085]), supporting Hypothesis 2.

In summary, Study 2 provides evidence for the mediating effect specified in the theoretical framework. However, the moderating role proposed in the model was not examined in Study 2. Study 3 was designed as a survey to assess the complete theoretical model, including the moderating effects. By integrating both experimental and field research methodologies, the overall research design is intended to enhance the external validity of the findings.

## 8. Study 3 Method

### 8.1. Participants and Procedures

For Study 3, full-time employees from the United States were recruited via Prolific, a platform widely used in academic research ([76]). Participants were assured that their responses would remain confidential, anonymous, and that the data would be used only for research purposes. To encourage thoughtful participation, each respondent received a compensation of 15 RMB (approximately 2.4 USD) for completing each wave of the survey. Following recommendations by [62] ([62]), attention check was included in every survey wave to ensure data quality. At Time 1, a total of 305 participants completed the initial survey, and 223 responses were received at Time 2, resulting in a retention rate of 73.1%. After excluding those who failed the attention check or submitted incomplete responses, the final sample included 218 participants. Among the final sample, 35.8% were female. Ethnically, 70.6% identified as Caucasian, 13.8% as African American, 9.2% as Asian, 3.2% as Hispanic American, and 2.8% did not specify. On average, participants had 16.37 years of education (SD = 2.86) and 7.49 years of organizational tenure (SD = 6.33).

### 8.2. Measures

To guarantee the precision of our survey materials, we followed standard translation and back-translation procedures ([11]).

AI dependence was assessed using the same three-item scale by [88] ([88]) employed in Studies 1 and 2 (Cronbach’s α = 0.79).

Employee’s personal sense of power toward the leader was measured with the eight-item scale from [7] ([7]), consistent with previous studies (Cronbach’s α = 0.91).

Employee voice behavior was evaluated using the three-item scale by [50] ([50]), as in earlier studies (Cronbach’s α = 0.87).

Coaching leadership was measured with an eight-item scale developed by [23] ([23]). Participants rated their agreement on a five-point scale (1 = strongly disagree, 5 = strongly agree). A sample item is: “My supervisor uses analogies, scenarios, and examples to help me learn.” The scale showed high reliability (Cronbach’s α = 0.89).

Control variables. Based on prior research indicating that demographic factors may influence voice behavior ([101]), we controlled for employee age. Additionally, since *self-efficacy* may correlate with both sense of power and voice behavior ([43]; [86]), we measured self-efficacy using a three-item scale developed by [84] ([84]) (e.g., “I am confident about my ability to do my job”; Cronbach’s α = 0.87).

### 8.3. Analytic Strategies

We initially evaluated the suitability of the four-factor model by Confirmatory Factor Analysis (CFA) in Mplus 8.3 and contrasted it with alternative models. Afterwards, we tested all hypotheses using PROCESS 3.0. To reduce multicollinearity among the variables, we centered employee AI dependence and coaching leadership ([77]). In addition, to improve the statistical power in detecting indirect effects, we used 95% confidence intervals (CIs).

## 9. Study 3 Results

### 9.1. Measurement Model Testing

Confirmatory factor analysis (CFA) was carried out using Mplus 8.3 software ([71]). As shown in Table 3, the proposed four-factor model, consisting of employee AI dependency, personal power towards leadership, employee voice behavior, and coaching leadership, exhibited an acceptable fit to the data (*χ*^2^ = 452.33, *df* = 203, CFI = 0.91, TLI = 0.89, RMSEA = 0.08, and SRMR = 0.06). These results support the discriminant validity ([38]).

### 9.2. Tests of Hypotheses

Descriptive statistics are detailed in Table 4. In line with our hypothesis, employee AI dependence was positively associated with employees’ personal sense of power toward their leaders (*b* = 0.16, *p* < 0.05), and this personal sense of power was also significantly linked to employee voice behavior (*b* = 0.45, *p* < 0.01).

Hypothesis 1 proposes a positive relationship between employee AI dependence and their sense of power toward their supervisor. As shown in Model 2 of Table 5, the regression results support this hypothesis, indicating a significant positive effect (*b* = 0.11, *p* < 0.05). Thus, Hypothesis 1 is supported. Hypothesis 2 propose that employee AI dependence positively influences voice behavior through the mediating role of their sense of power toward the leader. As shown in Model 5 of Table 5, sense of power had a significant positive effect on voice behavior (*b* = 0.36, *p* < 0.001). Bootstrapping analysis with 5000 resamples confirmed a significant indirect effect (*b* = 0.04, 95% Boot CI [0.0036, 0.0980]). Thus, these results support Hypothesis 2.

Hypothesis 3 proposes that coaching leadership strengthens the positive relationship between employee AI dependence and their sense of power toward the leader. As shown in Model 3 of Table 5, the interaction between coaching leadership and AI dependence was significant in predicting sense of power (*b* = 0.20, *p* < 0.001). Simple slope tests (Figure 2) reveal that under high levels of coaching leadership, AI dependence had a significant positive effect on sense of power (*b* = 0.13, *p* = 0.05). In contrast, under low coaching leadership, AI dependence was negatively related to sense of power (*b* = −0.16, 95% Boot CI [−0.29, −0.02], *p* = 0.02). This suggests that when leaders display fewer coaching behaviors, such as providing guidance and information, AI may be less able to substitute meaningfully for leadership, and reduced interaction with leaders could diminish employees’ sense of power. Thus, Hypothesis 3 is supported.

Hypothesis 4 states that coaching leadership moderates the indirect effect of employee AI dependence on voice behavior through sense of power. We used PROCESS Model 7 ([33]) to examine the conditional indirect effect of AI dependence on voice behavior via sense of power, as moderated by coaching leadership. A bootstrapping procedure with 20,000 resamples was applied to generate bias-corrected 95% confidence intervals (CIs). Results revealed a significant conditional indirect effect at high levels of coaching leadership (*b* = 0.05, 95% Boot CI [0.0015, 0.0966]), but not at low levels of coaching leadership (*b* = −0.06, 95% Boot CI [−0.1156, 0.0068]). The index of moderated mediation was significant (*index* = 0.07, 95% Boot CI [0.0131, 0.1267]), thereby supporting Hypothesis 4.

While Study 3 provided initial support for the model using a Prolific sample, Study 4 was designed to further assess the robustness and external validity of the findings. In Study 4, we recruited participants from organizational settings, which helps address concerns regarding the generalizability of results obtained from online platform data.

## 10. Study 4 Method

### 10.1. Participants and Procedures

In Study 4, participants were recruited through the executive development program (EDP) of a leading Chinese university. Working professionals enrolled in the EDP were invited to participate and encouraged to refer colleagues. This process generated a heterogeneous sample of 319 individuals from sectors such as services, training, manufacturing, and technology. Including participants from multiple industries helped enhance the generalizability of findings and reduced sector-specific bias, in line with recommendations by [78] ([78]). The Surveys were administered via WeChat along with a cover letter detailing the research purpose and ensuring confidentiality. Each respondent received a unique identifier to maintain anonymity while allowing responses to be matched across survey waves. Participants received 10 RMB (approximately USD 1.4) for each completed wave.

This study employs a three-wave longitudinal design. At Time 1 (T1), employees reported on coaching leadership, AI dependence, self-efficacy, and demographics. At Time 2 (T2), conducted one week later, employees’ personal sense of power in interactions with leaders was measured. Voice behavior was assessed at Time 3 (T3), one week after T2. Of the 319 initial participants, 285 completed all three waves, yielding a response rate of 89.34%. Participants who did not complete all three waves or who provided incomplete data were removed from the final dataset. The final sample consisted of 52.6% female participants, with an average age of 33.16 years (*SD* = 7.13), an average of 16.76 years of education (*SD* = 2.27), and an average supervisory tenure of 3.33 years (*SD* = 4.02).

### 10.2. Measures

The same measures from Study 3 were adopted in this study. Table 6 reported the reliability coefficients (Cronbach’s α) for all scales.

### 10.3. Analytic Strategies

We initially evaluated the suitability of the four-factor model by Confirmatory Factor Analysis (CFA) in Mplus 8.3 and compared it with several alternative models. We then tested all hypotheses using PROCESS 3.0. To reduce multicollinearity among the variables, we centered employee AI dependence, employee personal sense of power towards leader and coaching leadership ([77]). In addition, to improve the statistical power in detecting indirect effects, we used 95% confidence intervals (CIs).

## 11. Study 4 Results

### 11.1. Measurement Model Testing

Descriptive statistics are detailed in Table 6. To evaluate the measurement model, we employed Mplus version 8.3, a statistical software package used for estimating complex statistical models ([71]).

As shown in Table 7, the proposed four-factor model, which consists of employee AI dependency, personal power towards leadership, employee voice behavior, and coaching leadership, exhibited a satisfactory fit to the data (*χ*^2^ = 422.38, *df* = 203, *χ*^2^/df = 2.08; SRMR = 0.04, RMSEA = 0.06, CFI = 0.95; TLI = 0.95; [38]). which showed a better fit compared with the alternative model specifications.

### 11.2. Tests of Hypotheses

Supporting Hypothesis 1, the linear regression analysis shows a significant positive association between employee AI dependence and their sense of power toward the leader (*b* = 0.11, *p* < 0.05; see Table 8, Model 2). Results supported Hypothesis 2, indicating a significant positive effect of sense of power on voice behavior (*b* = 0.37, *p* < 0.001; see Table 8, Model 4). Furthermore, a bootstrapping procedure with 5000 resamples indicated a significant indirect effect of AI dependence on voice behavior through sense of power (*b* = 0.04, 95% Boot CI [0.0061, 0.0794]).

As shown in Model 3 of Table 8, the interaction between coaching leadership and employee AI dependence significantly predicted sense of power toward the leader (*b* = 0.11, *p* < 0.05). Simple slope tests (Figure 3) revealed that under high levels of coaching leadership, AI dependence had a significant positive effect on sense of power (*b* = 0.16, 95% Boot CI [0.0493, 0.2694], *p* < 0.01). In contrast, under low coaching leadership, AI dependence negatively influenced sense of power (*b* = −0.04, 95% Boot CI [−0.1834, 0.1017], *p* > 0.05). These findings suggest that higher levels of coaching leadership may enhance the extent to which AI dependence contributes to employees’ perceived power in leader–employee interactions. Thus, Hypothesis 3 is supported.

We use PROCESS Model 7 ([33]) to examine the conditional indirect effect of AI dependence on voice behavior via sense of power, moderated by coaching leadership. Bootstrapping with 20,000 resamples was applied to generate bias-corrected 95% confidence intervals (CIs). Results from Model 5 (Table 8) show a significant interaction between AI dependence and coaching leadership in predicting voice behavior (*b* = 0.15, *p* < 0.01). The conditional indirect effect of AI dependence on voice behavior through sense of power was significant when coaching leadership was high (*b* = 0.06, 95% Boot CI [0.0179, 0.1045]), but not significant when coaching leadership was low (*b* = −0.01, 95% Boot CI [−0.0841, 0.0387]). The index of moderated mediation was significant (*b* = 0.04, 95% Boot CI [0.0058, 0.0911]), further supporting Hypothesis 4.

## 12. Discussion and Conclusions

Drawing on Power Dependence Theory, this study examines how employees’ dependence on AI shapes their workplace voice behavior. Using two experimental studies and two surveys, the findings show that AI dependence enhances employees’ personal sense of power, which in turn moderates voice behavior. Furthermore, coaching leadership strengthens these relationships: the effects of AI dependence on both perceived power and voice behavior are more pronounced under high rather than low levels of coaching leadership. These results provide important theoretical and practical implications.

### 12.1. Theoretical Implications

This research makes several important theoretical contributions. First, this study uncovers a novel power dynamics mechanism through which AI dependence influences employee voice behavior. Traditionally, research has focused on static, hierarchical power dynamics within leader–subordinate dyads ([17]; [1]; [28]). In contrast, our research broadens the scope by examining the dynamic, triadic relationship among leaders, employees, and AI. As AI becomes more embedded in organizational processes ([35]), it reshapes power perceptions, enhancing employees’ sense of power and fostering voice behavior. By emphasizing power perceptions as a central mechanism, we offer a new theoretical explanation for how AI dependence drives employee voice, expanding on previous research that highlights employees’ reluctance to voice opinions due to power imbalances ([68]; [69]). This contribution shifts the focus from individual outcomes—such as autonomy, job insecurity, and performance ([2]; [58])—to the interpersonal dynamics within organizations, thus providing a more nuanced understanding of how AI not only influences individual work outcomes but also reshapes power dynamics and interpersonal interactions.

Second, this study explores the potential for AI to substitute certain leadership functions, particularly in the context of coaching leadership. By identifying coaching leadership as a key moderator in the relationship between AI dependence and perceived power, we address the question of whether AI can replace or complement specific leadership roles ([97]). Our findings suggest that AI can either substitute or enhance leadership functions, particularly those involving guidance and feedback. In this context, AI complements leadership behaviors, strengthening employees’ sense of power and altering traditional power dynamics. These insights contribute to ongoing debates regarding AI’s capacity to assume leadership responsibilities, adding depth to the literature on AI’s evolving role in leadership theory.

Finally, this study provides initial evidence that the impact of AI dependence transcends cultural boundaries. Despite the cultural differences between China and the United States, our results show consistent patterns: AI dependence positively influenced employees’ perceived power, which in turn promoted voice behavior. This suggests that AI’s impact on power dynamics and voice behavior reflects a broader, global shift in organizational structures, rather than being culturally specific. These findings challenge assumptions that voice behavior would differ significantly across cultures—particularly the expectation that collectivist cultures, such as China, would exhibit more restrained voice behavior ([39]; [66]). Thus, this study underscores the universal implications of AI in reshaping power perceptions and interpersonal relationships across diverse cultural contexts.

### 12.2. Practical Implications

Our study demonstrates that AI dependence enhances employees’ sense of power, thereby encouraging greater voice behavior in the workplace. However, this empowerment dynamic also presents challenges for leadership, as AI may strengthen employees’ confidence and initiative while potentially diminishing leaders’ perceived authority if not managed effectively. The following implications provide guidance on how organizations and leaders can respond to the changing landscape of AI-augmented work environments.

First, leadership remains crucial even as AI technologies evolve ([3]; [19]; [47]). As AI takes on more decision-support and feedback functions, leaders must develop complementary capabilities rather than compete with AI. Our findings suggest that coaching leadership—focused on guidance and feedback—aligns well with AI’s capabilities. Leaders should redefine their roles by emphasizing human connection, interpretation, and ethical judgment ([37]). By improving AI literacy and ethical awareness, leaders can ensure fairness and transparency in decision-making while maintaining human oversight.

Second, organizations must adopt a balanced approach to AI integration, acknowledging both its empowering potential and relational risks ([72]). AI should support, not replace, human leadership. Clear ethical boundaries are necessary, especially in performance evaluation and recruitment. Over-dependence on AI can reduce autonomy, creativity, and intrinsic motivation ([19]). Ethical frameworks should ensure AI empowers employees, rather than silences them. Furthermore, organizations should promote collaborative relationships and shared responsibility, positioning AI as a tool for collective development ([98]). Through thoughtful policy design and ongoing ethical review, AI and human leadership can work synergistically to foster employee voice and organizational vitality.

### 12.3. Limitations and Future Directions

Despite the valuable insights generated, several limitations in this research warrant attention and suggest directions for future studies.

First, our theoretical framework primarily draws on Power-Dependence Theory ([26]) to examine how AI reshapes leader–employee power dynamics. While this theory provides a strong foundation for understanding structural power shifts, it may not fully account for the motivational or relational processes involved in AI dependence. Future studies could integrate complementary perspectives, such as Self-Determination Theory (SDT), Social Exchange Theory (SET), or Algorithmic Management frameworks to offer a more holistic view of how AI affects intrinsic motivation, autonomy, and inter-personal trust. For instance, Self-Determination Theory may clarify whether AI’s impact on employees’ autonomy enhances or undermines intrinsic motivation and job satisfaction.

Second, although this study identifies employees’ sense of power as a key psycho-logical mechanism linking AI dependence to voice behavior, other pathways may also contribute to this relationship. AI dependence may alter affective or motivational states such as confidence, anxiety, or fatigue, which in turn influence employees’ willingness to speak up. Future research could examine interaction-related indicators, such as communication frequency or reliance on leader feedback, to determine whether increased AI dependence alters upward voice by reducing interpersonal reliance on supervisors. Beyond voice behavior, AI dependence may also shape a broader range of work behaviors, including creativity, risk-taking, or opportunism ([10]; [42]). Greater autonomy and information access may foster innovation, whereas excessive reliance on AI could potentially lead to isolation, diminished social connection, or self-serving tendencies ([32]). Exploring these dual consequences would enrich understanding of how AI dependence reshapes both social and task-oriented behaviors in organizations.

Third, although our experimental studies successfully isolated the core theoretical mechanisms of AI dependence, the simplified task scenarios may not fully capture the complexity of real-world power relations. The consulting-task setting captured AI guidance but lacked elements such as multi-stakeholder interactions, hierarchical tensions, or organizational politics, all of which are common in actual workplaces. Future research could adopt more realistic and context-rich designs to better reflect the multifaceted nature of power, decision-making, and influence in AI-enhanced environments. Combining experimental methods with field studies would also strengthen validity by balancing internal control with external realism.

Fourth, our reliance on self-report data in Studies 3 and 4 may have introduced common method variance (CMV), despite the use of time-lagged designs and controls for social desirability. Future research should employ multi-source data, such as supervisor ratings, peer assessments, or objective performance indicators and consider analytical approaches of unmeasured latent method factor (ULMF) techniques to mitigate CMV risk and strengthen causal inference.

Finally, cultural context likely plays a crucial role in shaping how employees perceive AI’s influence on power and voice. In high power-distance or collectivist cultures like China, employees may be less inclined to express voice or challenge authority due to hierarchical norms emphasizing respect and harmony ([36]; [27]; [39]). Future research should systematically compare cultural contexts to explore how national values, institutional systems, and leadership traditions moderate the effects of AI dependence. Such cross-cultural investigations would deepen our under-standing of how AI is embedded within different sociocultural systems and clarify its implications for leadership and employee agency across global workplaces.

## Figures and Tables

**Figure 1 behavsci-15-01709-f001:**
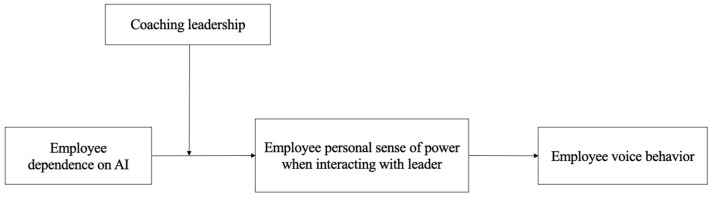
Proposed theoretical model.

**Figure 2 behavsci-15-01709-f002:**
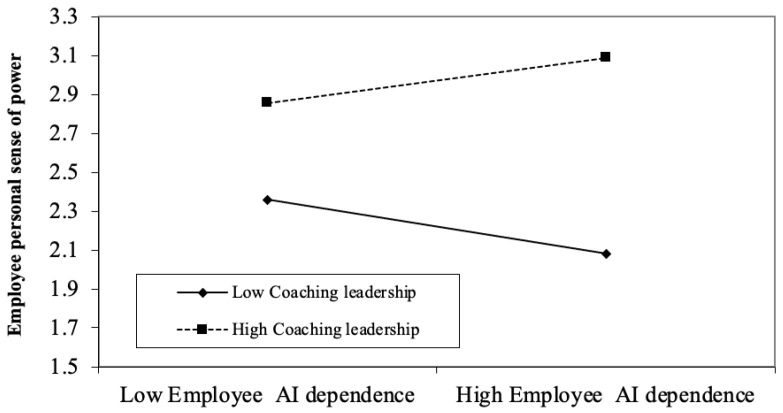
The Moderating Role of Coach Leadership in the Relationship between Employee AI Dependence and Employee’s Personal Sense of Power towards Leader in Study 3.

**Figure 3 behavsci-15-01709-f003:**
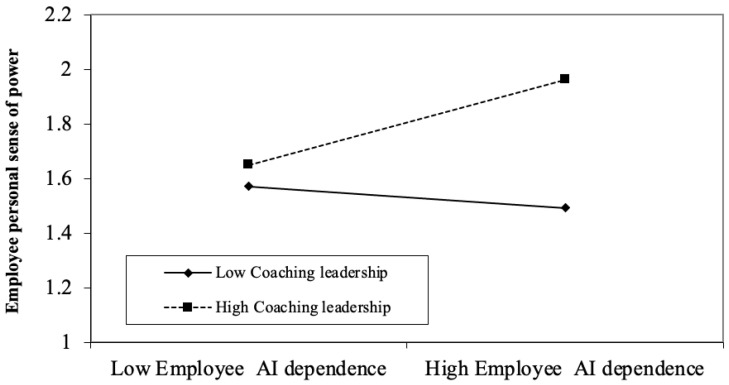
The Moderating Role of Coach Leadership in the Relationship between Employee AI Dependence and Employee’s Personal Sense of Power towards Leader in Study 4.

**Table 1 behavsci-15-01709-t001:** Means, Standard Deviations, and Correlations in Study 1.

Variable	Mean	SD	1	2	3
1. Condition (1 = dependence; 0 = control)	0.51	0.50	-		
2. AI dependence manipulation check	2.85	1.41	0.73 **	-	
3. Employee sense of power towards leader	3.39	0.66	0.16 *	0.06	-

Note. N = 172; * *p* < 0.05, ** *p* < 0.01.

**Table 2 behavsci-15-01709-t002:** Means, Standard Deviations, and Correlations in Study 2.

Variable	Mean	SD	1	2	3	4
1. Condition (1 = dependence; 0 = control)	0.50	0.50	-			
2. AI dependence manipulation check	3.04	1.24	0.85 **	-		
3. Employee sense of power towards leader	3.66	0.63	0.19 *	0.17 *	-	
4. Employee voice behavior	4.24	0.57	0.17 *	0.11	0.50 **	-

Note: N = 165; * *p* < 0.05, ** *p* < 0.01.

**Table 3 behavsci-15-01709-t003:** Results of Model Fit Estimates in Study 3.

Factors	*χ* ^2^	*df*	*χ*^2^/*df*	RMSEA	SRMR	CFI	TLI
Four factors	452.33	203	2.23 ***	0.08	0.06	0.91	0.89
Three factors. Personal sense of power and voice behavior combined	704.49	206	3.42 ***	0.11	0.08	0.81	0.79
Two factors. Personal sense of power, voice behavior and coach leadership combined	1159.31	208	5.57 ***	0.15	0.11	0.65	0.61
One factor. All variables combined	1310.53	209	6.27 ***	0.16	0.12	0.59	0.55

Note: N = 218; *** *p* < 0.001.

**Table 4 behavsci-15-01709-t004:** Descriptive Statistics and Interitem Correlations Among Variables in Study 3.

Variables	M	SD	1	2	3	4	5	6
1. Self-efficacy	4.28	0.65	-					
2. Age	40.81	10.31	0.15 *	-				
3. Employee AI dependence (T1)	3.35	0.88	0.03	−0.19 **	-			
4. Employee personal sense of power towards leader (T2)	3.52	0.75	0.35 **	−0.01	0.16 *	-		
5. Employee Voice behavior (T2)	3.66	0.79	0.43 **	0.15 *	0.12	0.45 **	-	
6. Coach leadership (T1)	3.66	0.74	0.27 **	−0.01	0.29 **	0.55 **	0.36 **	-

Note: N = 218; * *p* < 0.05, ** *p* < 0.01 (two-tailed tests).

**Table 5 behavsci-15-01709-t005:** Hypothesis Tests Results in Study 3.

Variables	Personal Sense of Power (T2)			Voice Behavior (T2)	
	Model 1	Model 2	Model 3	Model 4	Model 5
Constant	1.94 *** (0.35)	1.54 *** (0.40)	3.17 *** (0.84)	0.82 * (0.41)	0.28 (0.39)
Self-efficacy	0.40 *** (0.07)	0.39 *** (0.07)	0.19 ** (0.07)	0.50 *** (0.08)	0.36 (0.08)
Age	−0.01 (0.01)	−0.00 (0.00)	−0.00 (0.00)	0.01 (0.01)	0.01 * (0.01)
Employee AI dependence		0.11 * (0.06)	−0.73 ** (0.23)	0.11 * (0.06)	0.07 (0.05)
Coach leadership			0.54 *** (0.06)		
Employee AI dependence × Coach leadership			0.20 ** (0.06)		
Personal sense of power					0.36 *** (0.07)
*R* ^2^	0.13	0.15	0.36	0.21	0.31

Note: N = 218. Standard errors (SE) in parentheses. *** *p* < 0.001, ** *p* < 0.01, * *p* < 0.05.

**Table 6 behavsci-15-01709-t006:** Descriptive Statistics and Interitem Correlations Among Variables in Study 4.

Variables	M	SD	1	2	3	4	5	6
1. Age	33.16	7.13	-					
2. Self-efficacy	4.24	0.73	0.05	(0.91)				
3. Employee AI dependence (T1)	2.46	0.98	−0.16 **	−0.04	(0.89)			
4. Employee personal sense of power towards leader (T2)	3.25	0.79	0.02	0.33 **	0.12 *	(0.92)		
5. Employee Voice behavior (T2)	3.34	0.90	0.05	0.26 **	0.08	0.38 **	(0.91)	
6. Coach leadership (T1)	3.58	0.87	−0.12 *	0.10	0.21 **	0.19 **	0.01	(0.94)

Note: N = 285; Cronbach’s α in parentheses; * *p* < 0.05, ** *p* < 0.01(two-tailed tests).

**Table 7 behavsci-15-01709-t007:** Results of Model Fit Estimates in Study 4.

Factors	*χ* ^2^	*df*	*χ*^2^/*df*	RMSEA	SRMR	CFI	TLI
Four factors	422.38	203	2.08 ***	0.06	0.04	0.95	0.95
Three factors. Personal sense of power and voice behavior combined	932.85	206	4.53 ***	0.11	0.08	0.84	0.82
Two factors. Personal sense of power, voice behavior and coach leadership combined	2427.71	208	11.67 ***	0.19	0.22	0.52	0.46
One factor. All variables combined	2900.25	209	13.88 ***	0.21	0.23	0.41	0.35

Note: N = 285; *** *p* < 0.001.

**Table 8 behavsci-15-01709-t008:** Hypothesis Tests Results in Study 4.

Variables	Personal Sense of Power (T2)			Voice Behavior (T2)	
	Model 1	Model 2	Model 3	Model 4	Model 5
Constant	1.75 *** (0.33)	1.66 *** (0.33)	1.67 *** (0.32)	1.15 ** (0.38)	1.15 ** (0.38)
Self-efficacy	0.35 *** (0.06)	0.36 *** (0.06)	0.33 *** (0.06)	0.19 ** (0.07)	0.19 ** (0.07)
Age	0.00 (0.01)	0.00 (0.01)	0.00 (0.01)	0.01 (0.01)	0.01 (0.01)
Employee AI dependence		0.11 * (0.05)	0.06 (0.05)	0.05 (0.05)	0.03 (0.05)
Coach leadership			0.16 ** (0.05)		−0.04 (0.06)
Employee AI dependence × Coach leadership			0.11 * (0.05)		0.15 ** (0.06)
Personal sense of power				0.37 *** (0.07)	0.36 *** (0.07)
*R* ^2^	0.11	0.12	0.16	0.17	0.17

Note: N = 285. Standard errors (SE) in parentheses. *** *p* < 0.001, ** *p* < 0.01, * *p* < 0.05.

## Data Availability

The data that support the findings of this study are not openly available due to reasons of sensitivity and are available from the corresponding author upon reasonable request.

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
