# Peer review of "The Positive Effects of Employee AI Dependence on Voice Behavior—Based on Power Dependence Theory"

_behavsci, 2025, doi:10.3390/bs15121709_

Round 1

Reviewer 1 Report

Comments and Suggestions for Authors

Review for Behavior Science

The positive effects of employee AI dependence on Voice Behavior: Based on power dependence theory

Abstract

This paper focuses on the positive outcome of employee AI usage, suggesting that the higher level of dependency of AI gives rise to a higher level of power, which in turn, enhances their voice behavior. Furthermore, this research also examined the boundary condition of coaching leadership. Specifically, the more coaching leadership they received from their direct leader, employees' AI dependence has more voice at work via a personal sense of power over their leader. In general, the research discussed and resolved a common work behavior from a new theoretical perspective with strong empirical evidence. But, still, this study has some theoretical and logical issues that need to be further considered. I highlight the key ones below:

Theoretical concerns:

  1. I was wondering about the difference between AI usage and the AI dependence conceptually. More specifically, what unique behavior does the AI dependence explain beyond “AI usage” from the theoretical perspective? If not, why not follow the AI usage literature and invent a new word?

  1. According to the authors state “the decreased dependence diminishes perceived leader power and enhances employees’ sense of power in interactions with their supervisors? (I guess a typo here?).”, I also have a question about the logic. I agree with the former part of this argument, which may decrease the level of perceived leader power. However, why is the latter part? I don’t see the necessity of increasing one’s perceived power. The authors need more elaboration on this point.

  1. A follow-up question: why the personal sense of power enhanced by AI dependence uniquely worked “in interactions with leaders”, as suggested in H1? According to the discussion before the manuscript, employees are equipped with more information, more resources, or more professional suggestions. I take this as a kind of general resource that employees retrieved from AI. Why is the effect uniquely affected by interactions with leaders?

  1. A similar issue can also be found in the illustration of H2. The logic is relatively loose and lacks of strong argument to support such a hypothesis.

  1. Another issue is the choice of moderator in this study. The authors suggest that coaching leadership

Methodological concerns:

  1. The authors need to further explain the purpose of conducting four different studies. Especially for the relationship between S1 vs S2, S3 vs S4. For example, in S2, examining voice behavior by a self-rating scale in a lab condition cannot bring strong evidence of the effect on DV in this study. Therefore, I also suggest that the authors consider which study might be redundance in the manuscript.

  1. Why invite samples from different cultural backgrounds needs further elaboration.

  1. The figure is helpful for readers to understand the results in the experiment settings.

  1. The analysis strategy of H4 in S3 and S4 needs to be more detailed introduced, that is: how the moderated mediation is tested, by which method, etc.

In sum, this paper investigated an important research question and with robust results (multi-method, and samples from different cultures). The effect of AI usage at work needs more detailed investigation based on plenty of research results that show connections between using AI and productivity literature. Hence, I think the authors can further improve their hypothesis building and theoretical discussion in this manuscript.

Author Response

Comments 1: I was wondering about the difference between AI usage and the AI dependence conceptually. More specifically, what unique behavior does the AI dependence explain beyond “AI usage” from the theoretical perspective? If not, why not follow the AI usage literature and invent a new word?

Response 1: We sincerely appreciate this insightful comment. We fully agree that clarifying the conceptual and empirical distinctions between AI usage and AI dependence is essential for theoretical precision. In response, we have substantially revised the Introduction (p. 2) to clarify this differentiation.

First, conceptually, these two constructs capture distinct aspects of human–AI interaction. AI usage refers to the behavioral frequency and intensity of employees’ engagement with AI systems in daily work (Tang et al., 2022). It focuses on how often and to what extent employees use AI tools to perform tasks. In contrast, AI dependence is conceptualized as a psychological and situational state that describes the extent to which employees perceive that effective job performance requires reliance on AI’s advanced analytical and decision-making capabilities (Tang et al., 2023). This construct emphasizes the perceived necessity and irreplaceability of AI for accomplishing work goals, rather than mere behavioral engagement.

Second, the measurement of these constructs also differs. AI usage has been typically measured using items adapted from Medcof (1996), which focus on behavioral frequency (e.g., “I used AI to carry out most of my job functions”) (Tang et al., 2022). In contrast, AI dependence was measured using items inspired by Carr (2004), which capture employees’ perceived dependence on AI for essential decisions and task monitoring (e.g., “I depended on AI to make major work-related decisions”) (Tang et al., 2023). Thus, AI dependence captures a deeper cognitive and motivational dimension beyond behavioral use.

Finally, we have revised the introduction to explicitly reflect this conceptual distinction and justify why AI dependence provides additional theoretical insight beyond AI usage. Specifically, we position AI dependence as a state-like construct that may fluctuate across contexts and time (Shi et al., 2013; Shu et al., 2011), aligning with the notion of episodic dependence in social exchange and power-dependence frameworks. By distinguishing between usage and dependence, our study moves beyond adoption-level analysis to reveal how psychological reliance on AI reconfigures employees’ perceived power and voice behavior in leader–employee relationships.

Revised text (Introduction, p. 2, lines 44-73):

“Existing studies have primarily focused on the outcomes of AI usage, including its effects on performance, autonomy, innovation, and job security (Ackerman et al., 2020; Tang et al., 2022; Parker & Grote, 2022; Van Hootegem et al., 2019; Malik et al., 2022). AI usage refers to the extent to which employees interact with AI systems in their daily work (Tang et al., 2022), capturing the frequency and intensity of behavioral engagement with AI tools. However, as AI becomes more deeply embedded within organizational processes, employees may develop a more profound psychological reliance on intelligent systems. Recent developments illustrate that AI can even assume leadership functions—for instance, “Mika,” an AI system, was appointed as interim CEO at Dictador in 2022 (Mann, 2023; Höddinghaus et al., 2021). Yet, limited attention has been paid to how AI dependence influences the interpersonal dynamics between employees and human supervisors. Unlike usage, which focuses on behavioral engagement, AI dependence reflects a psychological and situational state that reshapes employees’ work experience rather than merely extending patterns of technological adoption.

Following Tang et al. (2023), AI dependence is defined as employees’ perception that effectively accomplishing their work requires reliance on AI’s advanced capabilities. In contrast to usage, which emphasizes how often AI is used, dependence emphasizes how essential and irreplaceable AI is perceived to be in enabling decision-making and task success. This conceptualization views AI dependence as a state-like and situational construct that may fluctuate across contexts and time, aligning with perspectives that treat dependence as episodic and context-specific (Shi et al., 2013; Shu et al., 2011). Accordingly, AI dependence captures a deeper level of human–technology interaction, offering new insight into how employees’ psychological perceptions and social relationships are re-configured through reliance on intelligent systems. Despite its growing importance, research on AI dependence remains scarce, particularly regarding its role in shaping leader–employee relationships. This study aims to fill this gap by examining how employees’ dependence on AI reshapes the power structure and interaction dynamics between managers and subordinates. By uncovering these evolving dynamics, organizations can better adapt leadership strategies to AI’s expanding role, thereby facilitating effective leadership transformation in the age of intelligent technology.”

These revisions clarify that AI dependence is not a linguistic variation of AI usage but a theoretically distinct construct that explains how dependence on AI as a critical resource alters employees’ perceived power and subsequent behaviors, consistent with Power Dependence Theory (Emerson, 1962).

Comments 2: According to the authors state “the decreased dependence diminishes perceived leader power and enhances employees’ sense of power in interactions with their supervisors? (I guess a typo here?).”, I also have a question about the logic. I agree with the former part of this argument, which may decrease the level of perceived leader power. However, why is the latter part? I don’t see the necessity of increasing one’s perceived power. The authors need more elaboration on this point. Response 2: We sincerely thank you for this insightful comment and the opportunity to clarify this theoretical mechanism. In the revised manuscript, we have expanded Section 2.2 (pp. 4-5) to provide a clearer explanation of why reduced dependence on leaders through AI integration can increase employees’ perceived power, rather than merely diminishing leader authority.

First, grounded in Power-Dependence Theory (Emerson, 1962), power within an exchange relationship is determined by the relative dependence between two parties. When an individual gains access to alternative resources that allow them to achieve valued goals independently, their dependence on the original power holder decreases, which in turn increases their relative power. Within organizational contexts, AI serves as such an alternative source of resources by providing employees with informational, decisional, and performance-related support traditionally controlled by leaders (Lee, 2018; Tsai et al., 2022). As employees gain greater control over task-relevant resources and reduce informational asymmetry, they occupy a stronger position in interpersonal exchanges, thereby enhancing their subjective sense of power (Aldrich & Herker, 1977; Spekman, 1979).

Second, as suggested by research on power (Keltner et al., 2003; Anderson & Galinsky, 2006; Sturm & Antonakis, 2015), an individual’s sense of power arises not only from hierarchical status but also from perceived autonomy and control over valued outcomes. Through AI-supported problem-solving and real-time feedback systems (Omoteso, 2012; Topol, 2019), employees can make independent decisions, manage resources, and evaluate their own performance without continuous leader intervention. This experience of self-sufficiency and autonomy fosters a heightened psychological sense of power in leader–employee interactions.

Accordingly, in the revised manuscript we emphasize that AI dependence does not represent passive dependence but rather an active empowerment process, whereby employees leverage AI as an alternative resource that enables greater independence, decision authority, and confidence in interactions with leaders. This mechanism aligns with the relational nature of power conceptualized by Emerson (1962)—as one party’s dependence decreases, its relative power and perceived control increase. We have revised Section 2.2 to reflect this clarification. The updated paragraph now reads (pp. 4-5, lines 177-208):

“When dependence decreases because alternative resources become available, the relative power of the previously dependent party increases (Emerson, 1962). In this sense, AI increasingly functions as an alternative source of leadership resources by performing managerial, informational, and cognitive tasks such as scheduling, task delegation, feedback provision, and performance monitoring (Lee, 2018; Wesche & Sonderegger, 2019; Tsai et al., 2022). By assisting employees in managing and executing work in ways comparable to human leaders (Gyory et al., 2022), AI reduces employees’ reliance on supervisors for direction, expertise, and feedback, thereby reconfiguring the asymmetrical power relationship that traditionally characterizes leader–follower interactions. This substitution is particularly evident in domains requiring information and expertise: AI provides accurate, objective, and comprehensive recommendations (Brynjolfsson & Mitchell, 2017; Jarrahi, 2018), mitigating the informational asymmetry that historically reinforces leaders’ authority (Maner & Mead, 2010). With transparent and accessible information provided by AI, employees gain greater control over task-relevant knowledge and decision-making resources. Consequently, as they acquire these alternative informational channels, they experience enhanced autonomy and occupy a stronger position in interpersonal exchanges with their leaders (Aldrich & Herker, 1977; Spekman, 1979).

Beyond informational substitution, AI enhances employees’ autonomy and efficacy in executing tasks. By supporting complex problem-solving and offering instantaneous, objective feedback (Omoteso, 2012; Topol, 2019; Wesche & Sonderegger, 2019), AI reduces the uncertainty and subjective bias often associated with human supervision (Höddinghaus et al., 2021). This expanded self-sufficiency enables employees to make independent decisions, manage resources, and evaluate outcomes without continuous leader intervention. Consistent with Power-Dependence Theory, as employees gain alternative means to achieve their goals, their relative power in the leader–employee relationship increases. Importantly, this enhancement of employees’ perceived power does not simply result from the erosion of leader authority but from employees’ heightened control over valued resources and their increased sense of autonomy and competence (Keltner et al., 2003; Anderson & Galinsky, 2006; Sturm & Antonakis, 2015). Hence, AI dependence reflects not passive reliance but an active empowerment process through which employees access alternative resources once monopolized by leaders, thereby strengthening their personal sense of power in interactions with leaders.”

Comments 3: A follow-up question: why the personal sense of power enhanced by AI dependence uniquely worked “in interactions with leaders”, as suggested in H1? According to the discussion before the manuscript, employees are equipped with more information, more resources, or more professional suggestions. I take this as a kind of general resource that employees retrieved from AI. Why is the effect uniquely affected by interactions with leaders?

Response 3: Thank you for this insightful follow-up question. We completely agree that AI provides employees with a broad range of benefits in terms of information, resources, and professional suggestions. However, our hypothesis specifically focuses on interactions with leaders because, as grounded in Power-Dependence Theory (Emerson, 1962), power is essentially relational, not a universal personal attribute. Power emerges within exchange relationships defined by mutual dependence, and in organizational contexts, this dependence is primarily vertical—employees rely on leaders for access to valuable resources such as information, guidance, decision authority, and performance feedback (Wee et al., 2017; Sturm & Antonakis, 2015).

AI dependence alters this vertical dependence structure by substituting leadership-controlled resources, such as task guidance, performance feedback, and expert advice (Lee, 2018; Wesche & Sonderegger, 2019; Tsai et al., 2022). As employees gain access to these alternative sources of guidance and expertise, their reliance on leaders decreases, leading to a reconfiguration of the asymmetrical power relationship between leaders and employees. Consequently, employees experience greater autonomy, control over information, and enhanced capabilities in their interactions with leaders, and this change in the power dynamic is most pronounced in these leader–employee interactions.

In contrast, relationships with colleagues or clients are typically not as hierarchically defined, nor are they based on a resource-dependent relationship. In these contexts, while AI may improve task efficiency or facilitate collaboration, it does not fundamentally alter interpersonal power dynamics. We have clarified this distinction in the revised Section 2.2 (pp. 4-5) to emphasize that the empowering effect of AI dependence is context-dependent—its most significant impact is seen in the leader–employee dyadic relationship.

We hope this elaboration clarifies the unique role of leadership interactions in the enhancement of employees’ sense of power due to AI dependence.

Comments 4: A similar issue can also be found in the illustration of H2. The logic is relatively loose and lacks of strong argument to support such a hypothesis.

Response 4: We sincerely appreciate this insightful comment regarding the theoretical logic underlying Hypothesis 2. We fully understand your concern that the initial version of our manuscript did not provide sufficiently strong theoretical reasoning to support why employees’ personal sense of power—enhanced by AI dependence—would lead to greater voice behavior. Accordingly, we have revised Section 2.3 (pp. 5–6, lines 215-246) to integrate these theoretical clarifications.

In the revised manuscript, we have substantially strengthened the theoretical justification for this hypothesis by more clearly grounding our arguments in Power Dependence Theory (Emerson, 1962) and expanding the explanation of how perceived power translates into proactive, risk-taking behaviors such as voice. Specifically, we now emphasize that power, according to this theory, emerges from asymmetric dependence relationships within social exchanges. When employees gain alternative access to valued resources through AI, their dependence on leaders decreases, which reduces relational constraints and expands behavioral freedom (Molm et al., 2000). This structural shift provides the foundation for employees to act with greater autonomy and assertiveness.

Furthermore, we elaborate that personal sense of power represents an internalized perception of one’s influence within a relationship (Keltner et al., 2003; Anderson et al., 2012). Prior research has shown that individuals who perceive themselves as powerful tend to behave more proactively and confidently, activating behavioral approach systems that promote initiative and goal pursuit (Bandura, 1999; Bugental & Lewis, 1999; Lin et al., 2019). We thus argue that the empowerment derived from reduced dependence on leaders through AI leads employees to feel more capable of influencing outcomes, which, in turn, encourages upward, change-oriented communication. In hierarchical relationships, those with low perceived power tend to remain silent due to fear of punishment, whereas employees with higher perceived power experience greater psychological safety and confidence to speak up (Morrison & Milliken, 2000; Ward et al., 2016; Liang et al., 2012).

The updated version more explicitly explains how AI-driven reductions in dependence translate into enhanced perceived power, and how this perceived power, in turn, facilitates voice behavior through increased behavioral freedom and confidence.

Revised 2.3 text (pp. 5–6, lines 215-246):

“According to Power Dependence Theory (Emerson, 1962), power arises from the asymmetrical dependence within social exchange relationships. When individuals gain alternative access to valuable resources, their dependence on others decreases, thereby reducing relational constraints and expanding their behavioral freedom. Specifically, individuals with relatively greater power are less constrained by external control and are more able to act according to their own judgment and goals (Molm et al., 2000). This change stems from structural shifts in dependence relationships, and consequently, in-dividuals’ behavior also changes.

Building on this, personal sense of power reflects an individual’s internal perception of their influence within the relationship (Keltner et al., 2003; Anderson et al., 2012). As Anderson et al. (2012) point out, individuals' beliefs about their power can shape their actual influence over others, beyond the effects of their sociostructural position. Those who perceive themselves as powerful tend to exhibit more confidence and proactive behaviors, which in turn activate their approach systems and encourage them to take initiative and pursue goals with greater autonomy (Bandura, 1999; Bugental & Lewis, 1999). Thus, employees with a stronger sense of power are more likely to see themselves as the "power-holder," gaining access to resources and opportunities necessary to achieve their goals (Lin et al., 2019).

In this context, employees' voice behavior is also significantly influenced by their personal sense of power. Voice behavior refers to employees’ discretionary upward communication aimed at improving work processes (Van Dyne & LePine, 1998; Detert & Burris, 2007), and it inherently involves a tendency to challenge existing norms and take risks. In hierarchical relationships, employees with lower perceived power are often silent for fear of disapproval or punishment (Morrison & Milliken, 2000; Ward et al., 2016). In contrast, employees who perceive themselves as having higher power (due to reduced dependence on leaders through AI) are more confident in expressing their thoughts and initiating constructive changes. As Liang et al. (2012) suggest, when employees perceive that they are more powerful, they infer that they have access to resources and opportunities to be effectively heard. Consequently, employees with a stronger sense of power are more likely to speak up in the workplace.”

Comments 5: Another issue is the choice of moderator in this study. The authors suggest that coaching leadership

Response 5:  We sincerely appreciate this thoughtful comment regarding our choice of coaching leadership as the moderating variable. We fully understand your concern about why this leadership style was selected and how it conceptually fits within the theoretical model. Following your valuable feedback, we have substantially strengthened the theoretical justification for this moderator in both the Introduction (p. 3) and Section 2.4 (pp. 6-7) of the revised manuscript.

First, we now more explicitly ground our reasoning in Power-Dependence Theory (Emerson, 1962), which emphasizes that power asymmetry between two parties depends on the control and substitutability of valued resources. Within organizations, leaders typically serve as the primary source of developmental guidance, performance feedback, and informational support. When alternative resource channels—such as AI systems—become available, employees’ dependence on leaders decreases, leading to a reconfiguration of power dynamics. Thus, we argue that leadership styles characterized by high levels of developmental and informational support are especially relevant for examining the substitutability of leadership functions by AI.

Second, we elaborate on why coaching leadership is theoretically distinct and particularly suitable for capturing this substitutability effect. Coaching leadership places a strong emphasis on supporting employees’ professional growth through open communication, constructive feedback, and empowerment (Eldor & Vigoda-Gadot, 2017; Mäkelä et al., 2024). It offers subordinates essential resources and informational guidance, helping them better understand organizational processes, align personal and organizational goals, and enhance engagement and motivation (Chughtai & Buckley, 2011; Yuan et al., 2019; Borde et al., 2024). These leadership functions are inherently resource-based, making coaching leadership a natural point of comparison to AI systems that perform similar roles—such as analyzing performance data, providing real-time feedback, and generating tailored skill-development suggestions (Brynjolfsson & Mitchell, 2017; Glikson & Woolley, 2020).

Third, we clarify that the functional overlap between AI and coaching leadership is central to our moderating hypothesis. When leaders demonstrate strong coaching behaviors, employees perceive AI as a viable alternative source of developmental and informational resources. This functional equivalence amplifies the impact of AI dependence on employees’ perceived power, as they gain access to resources traditionally mediated by their leaders. Conversely, under low coaching leadership, AI’s contributions are less substitutable for leadership functions, thus weakening its effect on power perceptions. In this way, coaching leadership does not merely act as a contextual boundary variable but captures the degree to which AI can substitute or complement human leadership in developmental domains.

Finally, we have revised Introduction (p. 3, lines 109-126) and Section 2.4 (pp. 6–7, lines 253-285) accordingly to integrate these theoretical clarifications and to more clearly articulate why coaching leadership provides a conceptually rich and empirically relevant boundary condition for our model.

Revised Introduction text (p. 3, lines 109-126):

“The substitutability of resource channels is crucial in understanding power dynamics (Emerson, 1962). Power-Dependence Theory suggests that the power imbalance between two parties is determined by their control over and access to valued resources. As AI systems increasingly take on roles traditionally performed by leaders—such as providing feedback, supporting decision-making, and facilitating skill development (Müller & Bostrom, 2016; Walsh, 2018)—employees' reliance on their leaders may decrease, leading to a shift in power dynamics. In this context, coaching leadership, which focuses on guiding professional development, providing feedback, and enhancing competence (Ali et al., 2020; Ladyshewsky & Taplin, 2017; Kellogg et al., 2020; Yuan et al., 2019), is particularly relevant due to its overlap with the capabilities of AI in providing developmental resources (Schildt, 2017). When coaching leadership is high, the functional overlap between leadership and AI is more pronounced, thereby amplifying the impact of AI dependence on employees' perceived power. In contrast, when coaching leadership is low, AI is less able to substitute for leadership functions, and the effect of AI dependence on perceived power is likely to be diminished. Thus, we propose that the degree of coaching leadership moderates the relationship between AI dependence and employees' perceived power, with the relationship being stronger when coaching leadership is high compared to when it is low.”

Revised 2.4 text (pp. 6-7, lines 253-285):

“According to Power-Dependence Theory (Emerson, 1962), power asymmetry between two parties depends on the control and substitutability of valued resources. When alternative sources become available, dependence on the original resource holder decreases, leading to a redistribution of power. Within organizations, leaders serve as a key source of valued resources—such as developmental guidance, performance feedback, and informational support—that employees rely on to accomplish their goals. Accordingly, when AI begins to replicate or substitute these leadership functions, employees’ dependence on leaders diminishes, enhancing their personal sense of power in interactions with leaders.

Among various leadership styles, coaching leadership is particularly relevant in this dynamic because of its strong focus on employee development and empowerment. Coaching leadership emphasizes supporting employees by establishing open communication channels, providing guidance, and inspiring growth (Eldor & Vigoda-Gadot, 2017; Mäkelä et al., 2024). It offers essential resources and informational support that enable employees to better understand organizational processes and develop greater control over their work environment, thereby boosting engagement and motivation (Chughtai & Buckley, 2011). Through problem-solving assistance, constructive feedback, and soliciting employee input, coaching leaders promote a mutually beneficial exchange that enhances employees’ competence and sustained vigor (Ely et al., 2010; Elloy, 2005; Yuan et al., 2019). Moreover, coaching leadership helps employees align personal and organizational goals and clarify role expectations (Borde et al., 2024), creating a developmental climate grounded in trust and autonomy.

However, these very functions—developmental feedback, informational guidance, and empowerment—are also the domains in which AI technologies have become increasingly capable. Modern AI systems can analyze performance data, offer real-time feedback, and generate tailored skill-enhancement suggestions (Brynjolfsson & Mitchell, 2017; Glikson & Woolley, 2020). When leaders demonstrate high levels of coaching leadership, employees perceive greater functional overlap between AI and their supervisors. This functional substitutability amplifies AI’s role as an alternative resource channel, further reducing employees’ dependence on leaders and enhancing their personal sense of power in leader–employee exchanges. Conversely, when coaching leadership is low, the overlap between AI’s capabilities and leadership behaviors is limited, making AI less likely to alter power dynamics.”

Comments 6: The authors need to further explain the purpose of conducting four different studies. Especially for the relationship between S1 vs S2, S3 vs S4. For example, in S2, examining voice behavior by a self-rating scale in a lab condition cannot bring strong evidence of the effect on DV in this study. Therefore, I also suggest that the authors consider which study might be redundance in the manuscript.

Response 6: We sincerely appreciate this insightful comment. We understand the reviewer’s concern regarding the rationale and necessity of conducting four separate studies, as well as the potential redundancy between Study 1 and Study 2, and between Study 3 and Study 4. In response, we have substantially clarified the distinct purpose, design logic, and contribution of each study in the revised “Overview of Studies” section (p. 7) to ensure the complementary role of all four studies is made explicit.

Specifically, our multi-study design was deliberately constructed to balance internal validity (causal inference through experiments) with external validity (ecological generalizability through surveys and field data), and to test the theoretical model across different methods, cultures, and data sources. The design logic follows a progressive validation sequence:

Study 1 (U.S. experiment) — Establishing causal evidence.

The first experiment was designed to establish the causal effect of AI dependence on employees’ perceived power (H1). Using experimental manipulation ensures high internal validity and tests the fundamental theoretical mechanism derived from Power-Dependence Theory.

Study 2 (China experiment) — Cross-cultural replication and mediation test.

Study 2 replicates the experimental logic in a different cultural context (China) to examine the cross-cultural robustness of the causal relationship identified in Study 1. Moreover, it extends the model by introducing voice behavior and mediation (H2), verifying whether the mechanism holds when behavioral outcomes are included.

We acknowledge that voice behavior was measured via self-report within a controlled setting; however, this design choice allowed us to preliminarily test the full theoretical pathway while maintaining experimental control. This step was critical before moving to field-based tests where causal inference is more limited.

Study 3 (U.S. two-wave survey) — Enhancing external validity.

Study 3 shifts from an experimental to a two-wave survey design to address concerns about generalizability and common method bias. Conducted with full-time U.S. employees over two time points, it provides evidence that the observed relationships persist in non-experimental, real-world data, thus strengthening the external validity of our findings.

Study 4 (China field study) — Contextual and ecological validation.

Finally, Study 4 validates the full theoretical model in an organizational field setting using a Chinese employee sample recruited from an executive development program. This study demonstrates that the effects of AI dependence are not limited to controlled or hypothetical scenarios but extend to natural workplace contexts, providing ecological validity and confirming the cross-cultural generalizability of the theoretical model.

We have revised the Overview of Studies section (p. 7, lines 304-326) to explicitly highlight these complementary purposes and the theoretical rationale for the sequential design, ensuring that no study appears redundant but rather contributes to the overall methodological rigor and theoretical robustness of the paper.

“To provide a comprehensive and rigorous test of our theoretical model, we conducted four complementary studies across different methods, samples, and cultural contexts. This multi-study design aimed to establish both the internal and external validity of our findings and to rule out methodological artifacts. Importantly, because the meanings of power and hierarchical dependence may vary across cultural contexts—being more pronounced in high power-distance societies such as China (Hofstede, 2001; Farh et al., 2007)—testing our model in both the United States and China allows us to assess whether the effects of AI dependence on power dynamics are universal or culturally contingent. Together, these studies form a progressive validation framework that integrates experimental control with field realism and cross-cultural verification. Study 1 employed an experimental design with a U.S. employee sample to establish the causal effect of AI dependence on employees’ perceived power (H1). Study 2, conducted in China, replicated this design but extended it by including the mediating and behavioral outcomes (H2). Although voice behavior was measured by self-report, this study served to verify the generalizability and cross-cultural robustness of the causal mechanism established in Study 1 and to preliminarily test the mediation path within a controlled setting. Study 3 adopted a two-wave survey among full-time employees in the United States to enhance external validity and demonstrate that the observed effects persist beyond experimental manipulation. Finally, Study 4, a field study conducted in China, further validated our theoretical model in a natural organizational context, supporting the real-world relevance and cross-cultural generalizability of the findings. Together, these studies integrate experimental control with field realism, demonstrating that the effects of AI dependence are robust across designs, measures, and cultural contexts.”

Comments 7: Why invite samples from different cultural backgrounds needs further elaboration.

Response 7: Thank you for your insightful comments and for your constructive feedback on our manuscript. We appreciate your suggestion to further elaborate on the rationale behind inviting samples from different cultural backgrounds.

In response, we have revised the Overview of Studies section to better explain why the inclusion of participants from both the United States and China is crucial for our study. Research on power dynamics (Hofstede, 2001; Farh et al., 2007) suggests that in high power-distance societies, such as China, hierarchical relationships and dependence on authority are more pronounced. In contrast, lower power-distance societies like the United States typically emphasize autonomy and egalitarianism in leader-employee relationships. By including samples from both cultures, our study can provide insights into whether the effects of AI dependence on power dynamics and voice behavior are universal or culturally contingent. This cross-cultural comparison is critical to understanding whether the empowering effects of AI are consistent across different cultural contexts or if they vary depending on societal norms surrounding authority and hierarchy.

Therefore, the inclusion of these two cultural contexts allows us to examine the generalizability of our findings while accounting for cultural differences in leadership and organizational behavior. This addition strengthens the external validity of our results and provides a more comprehensive understanding of the impact of AI dependence in different cultural settings.

We have updated the manuscript to reflect this reasoning more clearly and have provided further clarification in the Overview of Studies section (p. 7, lines 307-313).

“Importantly, because the meanings of power and hierarchical dependence may vary across cultural contexts—being more pronounced in high power-distance societies such as China (Hofstede, 2001; Farh et al., 2007)—testing our model in both the United States and China allows us to assess whether the effects of AI dependence on power dynamics are universal or culturally contingent. Together, these studies form a progressive validation framework that integrates experimental control with field realism and cross-cultural verification.”

Comments 8: The figure is helpful for readers to understand the results in the experiment settings.

Response 8: Thank you very much for your positive feedback. We are glad to hear that the figure effectively helps readers understand the results presented in our experimental studies. We appreciate your acknowledgment of its clarity and contribution to the paper’s readability.

If you have any additional suggestions on how the figure could be further improved—for example, in terms of layout, labeling, or clarity—we would be more than happy to make refinements accordingly.

Comments 9: The analysis strategy of H4 in S3 and S4 needs to be more detailed introduced, that is: how the moderated mediation is tested, by which method, etc.

Response 9: Thank you for your helpful comment regarding the need to provide more details about the analysis strategy for testing Hypothesis 4 in Studies 3 and 4. We appreciate your suggestion and have accordingly revised the manuscript to include a clearer and more detailed explanation of the moderated mediation testing procedure.

Specifically, we have now clarified that Hypothesis 4 was tested using PROCESS Macro Model 7 (Hayes, 2018), which allows examination of a conditional indirect effect—i.e., a mediation effect that is contingent on the level of a moderator. In both Studies 3 and 4, we used bootstrapping with 20,000 resamples to generate bias-corrected 95% confidence intervals (CIs) for the indirect effects at high and low levels of coaching leadership. The index of moderated mediation was also reported, as recommended by Hayes (2015), to statistically verify whether the indirect effect significantly varied across different levels of the moderator. We further clarified that the significance of the conditional indirect effects was determined based on whether the bootstrapped confidence intervals excluded zero. These details have been incorporated into the revised Results sections for both Study 3 and Study 4 (see p. 14 and 17).

Revised Study 3 text (p. 14, lines 566-572):

“We used PROCESS Model 7 (Hayes, 2018) to examine the conditional indirect effect of AI dependence on voice behavior via sense of power, moderated by coaching leadership. Bootstrapping with 20,000 resamples produced bias-corrected 95% confidence intervals (CIs). Results revealed a significant conditional indirect effect under high coaching leadership (b = 0.05, 95% Boot CI [0.0015, 0.0966]), but not under low coaching leadership (b = –0.06, 95% Boot CI [–0.1156, 0.0068]). The index of moderated mediation was significant (index = 0.07, 95% Boot CI [0.0131, 0.1267]), confirming Hypothesis 4.”

Revised Study 4 text (p. 17, lines 650-658):

“We used PROCESS Model 7 (Hayes, 2018) to examine the conditional indirect effect of AI dependence on voice behavior via sense of power, moderated by coaching leadership. Bootstrapping with 20,000 resamples produced bias-corrected 95% confidence intervals (CIs). Results from Model 5 (Table 8) indicated a significant interaction between AI dependence and coaching leadership in predicting voice behavior (b = 0.15, p < 0.01). The conditional indirect effect was significant under high coaching leadership (b = 0.06, 95% Boot CI [0.0179, 0.1045]) but not under low coaching leadership (b = –0.01, 95% Boot CI [–0.0841, 0.0387]). The index of moderated mediation was significant (b = 0.04, 95% Boot CI [0.0058, 0.0911]), further supporting Hypothesis 4.”

Comments 10: In sum, this paper investigated an important research question and with robust results (multi-method, and samples from different cultures). The effect of AI usage at work needs more detailed investigation based on plenty of research results that show connections between using AI and productivity literature. Hence, I think the authors can further improve their hypothesis building and theoretical discussion in this manuscript.

Response 10: We sincerely appreciate your thoughtful summary and encouraging evaluation of our work. We are grateful for your recognition of the importance of our research question and the robustness of our multi-method and cross-cultural design. At the same time, we fully acknowledge your valuable suggestion that the theoretical logic—especially the linkages between AI usage and productivity-related mechanisms—requires further refinement. In response, we have strengthened the hypothesis development by incorporating recent advances in the AI–productivity literature and clarified the theoretical pathways through which AI dependence influences employees’ psychological states and subsequent behaviors. We have also expanded the theoretical discussion in both the main text and the General Discussion to address the connections you identified. Thank you again for this insightful guidance—it has significantly improved the manuscript, and we remain willing to further refine as needed.

Reviewer 2 Report

Comments and Suggestions for Authors

It is a pleasure to review the article entitled “The Positive Effects of Employee AI Dependence on Voice Behavior--Based on power dependence theory.” This article presents a timely investigation the relationship between Employee AI Dependence and voice behavior from the power dependence perspective. Using two experimental studies and two survey investigations, the authors demonstrated that employee AI dependence is positively related to voice behavior via heightened perceptions of personal power, and that this relationship is strengthened under high levels of coaching leadership. The topic is interesting, the research design is rigorous, and the findings are relevant for both scholarship and practice. However, there are several limitations that need to be addressed, particularly in the areas of introduction, theoretical framework, and methods. Specific comments are listed below.

Introduction

1.A stronger theoretical rationale is needed to justify the exclusive focus on the relationship between employee AI dependence and voice behavior rather than other behaviors, such as creative problem-solving, taking charge, risk-taking behavior, expediency, or social loafing? :If AI enhances resources and autonomy, wouldn't a more direct outcome be improved job performance or creativity, rather than the socially risky act of voice? Additionally, Could AI dependence lead to less interaction with the leader, causing employees to become isolated and less likely to voice upward? The authors note that voice is a "non-compliant behavior" involving risk. However, they fail to convincingly argue why a shift in power dynamics caused by AI would primarily manifest as an increase in voice, as opposed to other behaviors.

Boussioux, L., Lane, J. N., Zhang, M., Jacimovic, V., & Lakhani, K. R. (2024). The crowdless future? Generative AI and creative problem-solving. Organization Science, 35(5), 1589-1607.

He, G., Yam, K. C., Zhao, P., Dong, X., Zheng, L., & Qin, X. (2025). Leaders inflate performance ratings for employees who use robots to augment their performance. Human Resource Management, 64(2), 543-563.

Hai, S., Long, T., Honora, A., Japutra, A., & Guo, T. (2025). The dark side of employee-generative AI collaboration in the workplace: An investigation on work alienation and employee expediency. International Journal of Information Management, 83, 102905.

Jia, N., Luo, X., Fang, Z., & Liao, C. (2024). When and how artificial intelligence augments employee creativity. Academy of Management Journal, 67(1), 5-32.

2. The coreassumption of the manuscript is that that an employee's dependence on AIincreases their personal sense of power toward the leader. Power dependence theory posits that power emerges as a function of one party’s dependence on the other for valued resources. An employee's dependence on AI signifies that AI holds power over the employee, not that the employee's own power is augmented. The more plausible mechanism would be that AI acts as an alternative resource, thereby reducing the employee’s dependence on the leader. The theoretical justification for selecting  sense of power as the key mediator requires further development.

3. I find it difficuly to understand why coaching leadership is susceptible to substitution by AI. AI operates on the data it is trained on. It struggles with the unique and political context of an organization—the "way things really work around here. A human coach understands organizational culture and interpersonal dynamics. They can help an employee connect their work to their personal life goals, values, and long-term career aspirations. This type of contextual and experiential wisdom is beyond the scope of current AI.

Method and results

4. Please provide more detailed information for your experimental scenarios in Study 1 and Study 2. The experimental scenariosare simplistic and lack the complexity of real-world organizational power dynamics. 

5. How can participants assess their sense of power toward the leader and frequence in engaging voice behavior in an imagined scenario? The authors used standardized scales (Anderson et al., 2012 for power; Lebel, 2016 for voice) but applied them to a hypothetical context. The instructions likely asked participants to imagine the scenario and then answer questions like:"In my interactions with my supervisor [in this scenario], I can get him/her to listen to what I say." and  "How frequently did you voice suggestions [in this task]?" The fundamental problem is that these questions are meaningless without a pre-existing, meaningful relationship with the "leader." In the scenario, the "leader" is not a real person. Participants have no history with this leader—no knowledge of their temperament, whether they are supportive or punitive, or how they typically react to suggestions. Their responses are essentially guesses based on a generic stereotype of a "boss," heavily influenced by their own past experiences or social desirability bias, not the experimental manipulation.

6. While a time-lagged design was usedin Study 3 and Study 4, all variables were still collected from the same source (employee self-reports). This introduces a high risk of common method variance artificially inflating the relationships between variables, particularly for perceptual constructs like sense of power and voice behavior. The authors should control for participants’ social desirability and employ the unmeasured latent method factor technique to address the concerns of common method variance.

Author Response

Comments 1: A stronger theoretical rationale is needed to justify the exclusive focus on the relationship between employee AI dependence and voice behavior rather than other behaviors, such as creative problem-solving, taking charge, risk-taking behavior, expediency, or social loafing? :If AI enhances resources and autonomy, wouldn't a more direct outcome be improved job performance or creativity, rather than the socially risky act of voice? Additionally, Could AI dependence lead to less interaction with the leader, causing employees to become isolated and less likely to voice upward? The authors note that voice is a "non-compliant behavior" involving risk. However, they fail to convincingly argue why a shift in power dynamics caused by AI would primarily manifest as an increase in voice, as opposed to other behaviors.

Response 1: Thank you for your insightful feedback and for raising a valuable concern regarding our exclusive focus on voice behavior as the primary outcome of AI dependence. We greatly appreciate the opportunity to clarify our rationale and strengthen the theoretical basis for this choice.

Firstly, we agree that AI dependence could potentially influence a variety of employee behaviors, including creative problem-solving, risk-taking, and task performance, as you mentioned. However, we argue that voice behavior is the most direct and pertinent outcome to examine in the context of AI’s impact on power dynamics for several reasons.

Power-Dependence Theory (Emerson, 1962) underscores that power is relational and shaped by interdependence. As AI provides employees with more control over critical resources, such as decision support, feedback, and task management, it reduces their reliance on leaders. This shift in power asymmetry manifests more clearly in behaviors related to interpersonal influence, which are integral to voice behavior. Specifically, voice involves challenging authority, proposing changes, and navigating social risks—activities that directly reflect power shifts in leader-employee relationships. In contrast, behaviors such as creative problem-solving or task performance are more aligned with resource utilization and individual capabilities, rather than the reconfiguration of relational dynamics. Therefore, we contend that voice behavior is a more appropriate measure of how AI affects the leader-employee power structure.

Second, regarding your concern about the potential negative effect of AI dependence—such as reduced interaction with leaders and isolation—we recognize this as a valid avenue for future research. While our study focuses on the empowering effect of AI on voice behavior, we acknowledge that reduced leader interaction could lead to disengagement or less willingness to voice concerns, potentially diminishing the need for upward communication. We have incorporated this idea into the limitations section of the manuscript and suggest that future research could explore how AI dependence influences employee autonomy and social connection with leaders.

Finally, these details have been incorporated into the revised Introduction section and Limitation and Future direction section (see pp. 2-3 and 20).

Revised Introduction text (pp. 2-3, lines 84-108):

“According to Power-Dependence Theory (Emerson, 1962), power arises from the dependency in social exchange relationships. When employees gain more resources through Artificial Intelligence (AI)—such as decision support, feedback, and skill enhancement—and reduce their dependence on leaders, their sense of power in interactions with leaders increases. The restructuring of power-dependence dynamics is often initially reflected in changes in interaction relationships (Molm, 1997), leading to adjustments in individuals' social interaction and influence behaviors (Anderson & Galinsky, 2006). Voice behavior refers to employees' voluntary communication aimed at suggesting ideas or recommendations to their supervisors to improve organizational operations (Morrison, 2023; Ng & Feldman, 2012; Van Dyne & LePine, 1998). This behavior is both constructive and socially risky, as it reflects employees' organizational commitment and motivation for improvement, but it can also be seen as a challenge to existing authority and decision-making (Morrison, 2011). In contrast, task-oriented behaviors, such as creativity or performance, reflect an individual’s use of resources and ability performance (Jia et al., 2024; Boussioux et al., 2024), rather than the redistribution of power within social relationships. Therefore, voice behavior is more capable of revealing the relational changes triggered by AI dependence—it reflects employees' repositioning and proactive influence tendencies in leader-employee relationships as their sense of power increases (Tost et al., 2013). When employees perceive higher power, they are more willing to take social risks and proactively voice their opinions, thus promoting organizational improvement (Ng & Feldman, 2012; Chamberlin et al., 2017; Morrison, 2023; Luo et al., 2024). By providing reliable information and feedback, AI partially substitutes for leadership functions, altering the structure of employees' dependence on leaders, enabling them to gain greater autonomy and control over information in interactions with leaders, which in turn enhances their sense of power and stimulates more proactive voice behavior.”

Revised Limitation and Future direction text (p. 20, lines 789-802):

“Second, although this study identifies employees’ sense of power as the central psychological mechanism linking AI dependence to voice behavior, alternative pathways may also explain this relationship. AI dependence could alter affective or motivational states such as confidence, anxiety, or fatigue, which in turn influence employees’ willingness to speak up. Future research may examine indicators of leader–employee interaction (e.g., communication frequency, feedback reliance) to determine whether heightened AI dependence reduces upward voice by weakening interpersonal reliance on leaders. Furthermore, while this study focuses on voice behavior as a key relational outcome, AI dependence may also shape other behaviors—such as creativity, risk-taking, or opportunism (Boussioux et al., 2024; Jia et al., 2024). Greater autonomy and information access might foster innovation, whereas excessive reliance on AI could generate isolation, alienation, or self-serving tendencies (Hai et al., 2025). Exploring these dual consequences would enrich understanding of how AI dependence reshapes both social and task-oriented behaviors in organizations.”

Comments 2: The coreassumption of the manuscript is that an employee's dependence on AI increases their personal sense of power toward the leader. Power dependence theory posits that power emerges as a function of one party’s dependence on the other for valued resources. An employee's dependence on AI signifies that AI holds power over the employee, not that the employee's own power is augmented. The more plausible mechanism would be that AI acts as an alternative resource, thereby reducing the employee’s dependence on the leader. The theoretical justification for selecting sense of power as the key mediator requires further development.

Response 2: Thank you for your thoughtful comments and suggestions. We greatly appreciate the opportunity to clarify our theoretical reasoning and improve the manuscript based on your feedback.

First, in response to your valuable suggestion, we have refined our explanation of the mediating role of employees' sense of power in the relationship between AI dependence and voice behavior. According to Power-Dependence Theory (Emerson, 1962), power is inherently relational, arising from dependence dynamics. When employees gain access to alternative resources—such as decision support, feedback, and skill development through AI—their reliance on leaders diminishes, which in turn enhances their perceived power. This shift in power dynamics plays a crucial role in understanding how AI dependence affects employees' behavior, particularly in social exchanges where the ability to influence others becomes more pronounced.

Second, we have clarified that the change in perceived power resulting from AI dependence is not simply a passive reliance on technology, but an empowerment process. By providing employees with greater autonomy, control over task-related knowledge, and decision-making capabilities, AI enables them to engage more assertively in organizational communication (Keltner et al., 2003; Anderson & Galinsky, 2006). This empowerment is directly linked to voice behavior, as employees feel more confident and capable of initiating constructive changes (Ng & Feldman, 2012; Chamberlin et al., 2017).

Third, to address your comment on the theoretical justification for sense of power as a mediator, we have further developed our argument that voice behavior is a more suitable outcome for exploring the relational shifts caused by AI dependence. Voice behavior, inherently tied to social interactions and challenges to authority, offers a clearer manifestation of the changes in power perception triggered by AI

Finally, as per your suggestion, we have revised the Theory and Hypotheses section (see pp. 5-6).

Revised Introduction text (pp. 5-6, lines 215-246):

“According to Power Dependence Theory (Emerson, 1962), power arises from the asymmetrical dependence within social exchange relationships. When individuals gain alternative access to valuable resources, their dependence on others decreases, thereby reducing relational constraints and expanding their behavioral freedom. Specifically, individuals with relatively greater power are less constrained by external control and are more able to act according to their own judgment and goals (Molm et al., 2000). This change stems from structural shifts in dependence relationships, and consequently, in-dividuals’ behavior also changes.

Building on this, personal sense of power reflects an individual’s internal perception of their influence within the relationship (Keltner et al., 2003; Anderson et al., 2012). As Anderson et al. (2012) point out, individuals' beliefs about their power can shape their actual influence over others, beyond the effects of their sociostructural position. Those who perceive themselves as powerful tend to exhibit more confidence and proactive behaviors, which in turn activate their approach systems and encourage them to take initiative and pursue goals with greater autonomy (Bandura, 1999; Bugental & Lewis, 1999). Thus, employees with a stronger sense of power are more likely to see themselves as the "power-holder," gaining access to resources and opportunities necessary to achieve their goals (Lin et al., 2019).

In this context, employees' voice behavior is also significantly influenced by their personal sense of power. Voice behavior refers to employees’ discretionary upward communication aimed at improving work processes (Van Dyne & LePine, 1998; Detert & Burris, 2007), and it inherently involves a tendency to challenge existing norms and take risks. In hierarchical relationships, employees with lower perceived power are often silent for fear of disapproval or punishment (Morrison & Milliken, 2000; Ward et al., 2016). In contrast, employees who perceive themselves as having higher power (due to reduced dependence on leaders through AI) are more confident in expressing their thoughts and initiating constructive changes. As Liang et al. (2012) suggest, when employees perceive that they are more powerful, they infer that they have access to resources and opportunities to be effectively heard. Consequently, employees with a stronger sense of power are more likely to speak up in the workplace.”

Comments 3: I find it difficulty to understand why coaching leadership is susceptible to substitution by AI. AI operates on the data it is trained on. It struggles with the unique and political context of an organization—the "way things really work around here. A human coach understands organizational culture and interpersonal dynamics. They can help an employee connect their work to their personal life goals, values, and long-term career aspirations. This type of contextual and experiential wisdom is beyond the scope of current AI.

Response 3: Thank you for your thoughtful feedback and valuable suggestions. We greatly appreciate the opportunity to revise our manuscript, and we have carefully addressed your concerns to strengthen the clarity and rigor of the paper.

First, we appreciate your comment regarding the susceptibility of coaching leadership to substitution by AI. We agree that AI, while increasingly capable of analyzing data and providing feedback, lacks the unique ability of human leaders to navigate the political and social dynamics of an organization. Specifically, AI struggles to account for the nuances of organizational culture, interpersonal relationships, and long-term career goals—elements central to coaching leadership. However, our argument is that coaching leadership and AI functions overlap in certain domains, especially in areas such as feedback provision, skill development, and task guidance (Brynjolfsson & Mitchell, 2017; Glikson & Woolley, 2020). When coaching leadership is high, the functional overlap between AI and leadership behaviors increases, making employees feel more empowered and engaged in upward voice behaviors (Van Dyne & LePine, 1998; Detert & Burris, 2007). We further elaborate on this point in the Theory and Hypotheses section (p. 6).

Second, following your suggestion, we have expanded our discussion in the practical implications section to emphasize the continuing importance of human leadership (p. 19), even in an AI-driven environment. While AI is increasingly taking on analytical, feedback, and decision-support roles, the human element in leadership remains essential. Leaders must adapt by enhancing their capacity to complement AI technologies, focusing on fostering human connection, ethical judgment, and interpretation rather than competing with AI. We have emphasized that leaders with strong coaching capabilities will be in a better position to integrate AI into their leadership style, ensuring that AI acts as a supportive tool that enhances human judgment and decision-making.

Revised Theory and Hypotheses text (p. 6, lines 253-285):

“According to Power-Dependence Theory (Emerson, 1962), power asymmetry between two parties depends on the control and substitutability of valued resources. When alternative sources become available, dependence on the original resource holder decreases, leading to a redistribution of power. Within organizations, leaders serve as a key source of valued resources—such as developmental guidance, performance feedback, and informational support—that employees rely on to accomplish their goals. Accordingly, when AI begins to replicate or substitute these leadership functions, employees’ dependence on leaders diminishes, enhancing their personal sense of power in interactions with leaders.

Among various leadership styles, coaching leadership is particularly relevant in this dynamic because of its strong focus on employee development and empowerment. Coaching leadership emphasizes supporting employees by establishing open communication channels, providing guidance, and inspiring growth (Eldor & Vigoda-Gadot, 2017; Mäkelä et al., 2024). It offers essential resources and informational support that enable employees to better understand organizational processes and develop greater control over their work environment, thereby boosting engagement and motivation (Chughtai & Buckley, 2011). Through problem-solving assistance, constructive feedback, and soliciting employee input, coaching leaders promote a mutually beneficial exchange that enhances employees’ competence and sustained vigor (Ely et al., 2010; Elloy, 2005; Yuan et al., 2019). Moreover, coaching leadership helps employees align personal and organizational goals and clarify role expectations (Borde et al., 2024), creating a developmental climate grounded in trust and autonomy.

However, these very functions—developmental feedback, informational guidance, and empowerment—are also the domains in which AI technologies have become increasingly capable. Modern AI systems can analyze performance data, offer real-time feedback, and generate tailored skill-enhancement suggestions (Brynjolfsson & Mitchell, 2017; Glikson & Woolley, 2020). When leaders demonstrate high levels of coaching leadership, employees perceive greater functional overlap between AI and their supervisors. This functional substitutability amplifies AI’s role as an alternative resource channel, further reducing employees’ dependence on leaders and enhancing their personal sense of power in leader–employee exchanges. Conversely, when coaching leadership is low, the overlap between AI’s capabilities and leadership behaviors is limited, making AI less likely to alter power dynamics.”

Revised Practical Implications text (p. 19, lines 738-750):

“First, even in the era of advanced AI applications, leadership remains indispensable (Agrawal et al., 2017; Davenport & Kirby, 2016; Kolbjørnsrud et al., 2017; Wilson & Daugherty, 2018). As AI increasingly undertakes decision-support and feedback functions, leaders must adapt by developing capabilities that complement, rather than compete with, intelligent systems. Our findings suggest that when leaders display strong coaching leadership behaviors, their functions often overlap with AI’s analytical and feedback capabilities. This overlap calls for leaders to redefine their roles—placing less emphasis on task control and more on fostering human connection, interpretation, and ethical judgment. Leading responsibly in AI-augmented contexts requires maintaining human oversight and accountability across all stages of AI application (Hossain et al., 2025). By improving AI literacy and ethical awareness, leaders can ensure fairness, transparency, and auditability in algorithmic decision-making, guarding against biases and promoting inclusive and equitable outcomes.”

Comments 4: Please provide more detailed information for your experimental scenarios in Study 1 and Study 2. The experimental scenarios are simplistic and lack the complexity of real-world organizational power dynamics. 

Response 4: Thank you for your valuable comments and insightful suggestions. We appreciate your feedback and have made revisions to address the concerns you raised.

First, we acknowledge that the original description of our experimental scenarios was relatively brief. In the revised manuscript, we have provided more detailed information about the experimental setup (see Section 4.2 p. 8) and clarified that our design followed the experimental framework developed by Tang et al. (2023), which has been widely used in the literature for empirical research on AI dependence in controlled environments.

Specifically, participants were asked to imagine themselves as consultants at a management consulting firm, advising a client on strategic decisions for a lemonade business. In the AI dependence condition, participants interacted with an intelligent algorithm through a visualized conversational interface embedded in the survey, receiving data-driven recommendations and feedback (such as insights into customer satisfaction and profitability). In the Control condition, participants made the same decisions independently, without the aid of AI.

Second, we also addressed your comment in the Limitations and Future Research section (see Section 12.3 p. 20), where we note that future studies could expand upon our experimental design by using more diverse work context simulations or field experiments. This would allow for a deeper exploration of the interpersonal relationships and political dynamics that are inherent in real-world organizational power structures.

Revised Manipulations text (p. 8, lines 359-374):

“AI dependence. Following the experimental framework developed by Tang et al. (2023), participants were asked to imagine themselves as consultants in a management consulting firm, responsible for advising a client on strategic decisions for a new lemonade business.

In the AI dependence condition (n = 87), participants collaborated with an intelligent algorithm that interacted with them through a series of visualized conversational interfaces embedded in the survey. The algorithm provided data-driven recommendations and feedback—for instance, highlighting how variations in sugar, lemon, and color levels would affect customer satisfaction and profitability—and in some cases explicitly indicated when a participant’s choice was suboptimal based on its analytical knowledge. These interactions required participants to depend on the AI’s informational input to make informed recommendations.

In the control condition (n = 85), participants completed the same decision-making task independently, making all recommendations based on their own reasoning and knowledge without algorithmic assistance. The two conditions were otherwise identical in task structure, decision sequence, and informational content.”

Revised Limitations and Future Directions text (p. 20, lines 803-811):

“Third, although our experimental studies successfully isolated the core theoretical mechanisms of AI dependence, the simplified task scenarios may not fully capture the complexity of real-world power relations. The consulting-task setting simulated AI guidance but lacked elements such as multi-stakeholder interactions, hierarchical tensions, and organizational politics that characterize actual workplaces. Future research could adopt more realistic and context-rich designs to better reflect the multifaceted nature of power, decision-making, and influence in AI-enhanced environments. To further strengthen validity, subsequent studies should combine experimental and field designs to bridge internal and external validity.”

Comments 5: How can participants assess their sense of power toward the leader and frequence in engaging voice behavior in an imagined scenario? The authors used standardized scales (Anderson et al., 2012 for power; Lebel, 2016 for voice) but applied them to a hypothetical context. The instructions likely asked participants to imagine the scenario and then answer questions like:"In my interactions with my supervisor [in this scenario], I can get him/her to listen to what I say." and  "How frequently did you voice suggestions [in this task]?" The fundamental problem is that these questions are meaningless without a pre-existing, meaningful relationship with the "leader." In the scenario, the "leader" is not a real person. Participants have no history with this leader—no knowledge of their temperament, whether they are supportive or punitive, or how they typically react to suggestions. Their responses are essentially guesses based on a generic stereotype of a "boss," heavily influenced by their own past experiences or social desirability bias, not the experimental manipulation.

Response 5: Thank you very much for this insightful comment. We fully understand the concern regarding the use of standardized measures of sense of power (Anderson et al., 2012) and voice behavior (Lebel, 2016) within an imagined scenario where participants have no pre-existing relationship with a leader. We have carefully revised both the experimental manipulation (Section 4.2) and the measurement procedures (Section 6.3) to clarify how the study ensured contextual immersion and minimized the risk of abstract or stereotypical responses.

First, we acknowledge that the original description of the experimental scenario was too brief. In the revised manuscript, we added detailed information explaining that participants were asked to simulate a consulting interaction by providing an open-ended response following the AI dependence manipulation. Specifically, after receiving AI-generated feedback, participants were instructed to:

“Please explain the logic behind your consulting recommendation and use it as supporting material to convince your supervisor to adopt your proposal (at least 50 words).”

This task required participants to mentally simulate an authentic leader–employee exchange, helping them frame their responses to subsequent scales based on the experimental manipulation rather than on generic assumptions about “a boss.” (see Section 4.2, p. 8, lines 365-371).

Second, we clarified that explicit scenario-based instructions were given prior to each measurement scale. For example, before assessing sense of power, participants were told to “evaluate how you would feel in your interactions with your supervisor in the scenario you just experienced,” and before rating voice behavior, they were instructed to “assess how likely you would be to engage in the following behaviors in this scenario.” These instructions ensured that participants’ responses reflected their reactions within the task context, not their real-life workplace experiences (see Section 6.3, p. 10, lines 425-430).

Comments 6: While a time-lagged design was used in Study 3 and Study 4, all variables were still collected from the same source (employee self-reports). This introduces a high risk of common method variance artificially inflating the relationships between variables, particularly for perceptual constructs like sense of power and voice behavior. The authors should control for participants’ social desirability and employ the unmeasured latent method factor technique to address the concerns of common method variance.

Response 6: Thank you very much for your insightful comments and for raising the important issue of common method variance (CMV). We fully understand your concern, especially considering that Studies 3 and 4 utilized self-report data from a single source, which can introduce a risk of CMV potentially inflating the relationships between variables such as sense of power and voice behavior.

First, to address the potential CMV issue, we performed a Harman’s single-factor test in both Study 3 and Study 4. The results showed that, in Study 3, the first factor explained 34.8% of the variance, and in Study 4, the first factor explained 28.9% of the variance. According to the standard for Harman’s test, if the first factor accounts for more than 40% of the variance, it typically indicates significant CMV. However, in our case, the first factor explained less than 40% of the variance, and there were multiple factors, suggesting that CMV is unlikely to have had a substantial effect on our results. This finding further supports the conclusion that CMV has minimal impact on the outcomes of our study.

Second, we acknowledge the potential impact of social desirability bias on self-reported data, particularly when measuring sensitive constructs like sense of power and voice behavior. To mitigate this bias, we included a detailed informed consent form at the beginning of the survey, clearly informing participants that their responses would be kept confidential. This was done to reduce any pressure participants may have felt to answer in a socially desirable manner. Furthermore, we carefully designed the survey questions with neutral phrasing to avoid leading participants, ensuring that their responses were as authentic and unbiased as possible.

Lastly, although we did not employ the unmeasured latent method factor (ULMF) technique in the current study, we greatly appreciate your constructive suggestion and will consider incorporating this method in future research. The ULMF technique is an effective approach to differentiate the bias introduced by measurement methods from the true relationships between variables. We believe that employing this method would further strengthen the reliability of the findings and ensure that the conclusions drawn are not impacted by CMV. We have also noted this in the Limitations and Future Directions section (see p. 20, lines 812-817), where we discuss plans to include ULMF in future studies.

Revised Limitations and Future Directions text (p. 20, lines 812-817):

“Fourth, our reliance on self-report data in Studies 3 and 4 may have introduced common method variance (CMV), despite the use of time-lagged designs and controls for social desirability. Future research should employ multi-source data (e.g., supervisor evaluations, peer assessments, or objective performance indicators) and consider the use of unmeasured latent method factor (ULMF) techniques to mitigate CMV risk and strengthen causal inference.”

Once again, we thank you for your valuable suggestions, and we will continue to refine our methods in future research to improve the robustness and reliability of our results.

Reviewer 3 Report

Comments and Suggestions for Authors

Dear Authors,

I would like to thank you for the opportunity to review this manuscript. The paper addresses a highly timely and scientifically relevant topic: the reconfiguration of organizational power in the context of Artificial Intelligence (AI) integration. The connection established between Power Dependence Theory and employee voice behavior constitutes a valuable and relatively unexplored theoretical contribution. The finding that coaching leadership amplifies the positive effects of AI dependence on perceived power and voice behavior is particularly insightful, suggesting that AI can complement (but not replace) leadership functions oriented toward development and autonomy. This supports the emerging paradigm of AI-augmented leadership rather than AI-substituted leadership.

The introduction effectively situates the research within the broader literature, though it could benefit from incorporating complementary frameworks such as Self-Determination Theory, Social Exchange Theory, or Algorithmic Management perspectives, which would enrich the conceptual grounding and reinforce the manuscript’s theoretical positioning vis-à-vis convergent studies.

The references are recent and relevant, combining classical theoretical sources with empirical research from 2020–2024. Nonetheless, citation density could be optimized in Section 2.1, where certain classical works are reiterated without adding new interpretive depth. Similarly, the empirical justification for selecting coaching leadership as the moderating variable could be expanded to strengthen its conceptual distinctiveness.

The methodological design, a sequential program of four studies combining experimental and survey-based approaches (two in the U.S. and two in China), demonstrates a robust and coherent framework consistent with contemporary standards in experimental organizational research. This mixed design effectively reinforces both internal and external validity, as well as cross-cultural replicability of the theoretical model.

There remain, however, several areas for refinement:

  1. The criteria for participant exclusion and the randomization procedures could be more explicitly described.
  2. The discussion could briefly address the potential influence of cultural context (Chinese sample) on both voice expression and power perception, as these constructs are sensitive to cultural dimensions such as collectivism.
  3. Although the study achieves a high level of ecological validity, a more explicit reflection on institutional and hierarchical factors (for example, vertical communication and status orientation in Chinese workplaces) would further contextualize the findings.

The discussion and practical implications are written with conceptual maturity and intellectual caution. The manuscript avoids technological determinism and presents a balanced, human-centered view of leadership transformation in the AI era. The call for prudence and leader adaptation is well substantiated. Nonetheless, the paper could be further strengthened by including more actionable recommendations, such as guidelines for AI policy design in HRM, leadership development programs, or ethical boundary management in algorithmic decision-making.

Overall, this is a well-conceived, empirically grounded, and theoretically original contribution. With modest revisions aimed at deepening theoretical integration and refining cultural contextualization, the paper would be well positioned for publication in a leading journal like Behavioral Sciences.

I commend the authors for their rigorous work and wish them continued success in their research endeavors.

Sincerely,

Author Response

Comments 1: The introduction effectively situates the research within the broader literature, though it could benefit from incorporating complementary frameworks such as Self-Determination Theory, Social Exchange Theory, or Algorithmic Management perspectives, which would enrich the conceptual grounding and reinforce the manuscript’s theoretical positioning vis-à-vis convergent studies.

Response 1: We sincerely thank the reviewer for their insightful comments and for suggesting the incorporation of complementary frameworks such as Self-Determination Theory (SDT), Social Exchange Theory (SET), and Algorithmic Management perspectives. While we recognize the value of these frameworks in providing additional depth to the study of AI's impact on employee behavior, we have chosen to maintain our focus on Power-Dependence Theory (Emerson, 1962) as the central theoretical framework for understanding the dynamics between AI dependence and employee behavior.

Power-Dependence Theory offers a comprehensive foundation for examining how changes in resource exchange relationships between leaders and employees influence their interactions. The theory’s emphasis on how power arises from mutual dependence in social exchange relationships aligns well with our focus on AI as a new source of resources, reducing employees' reliance on leaders, and thereby reshaping the power dynamics within these relationships. This framework enables us to specifically address how AI alters the traditional power hierarchy, empowering employees in their interactions with leaders.

While we did not incorporate the additional frameworks suggested by the reviewer, we believe that the integration of Power-Dependence Theory with our specific focus on AI dependence and voice behavior provides a robust and coherent explanation for the observed effects. This theoretical position allows us to explore the psychological and situational shifts that AI dependence triggers, without diluting the focus on power dynamics between leaders and employees.

In response to your comment, we have further elaborated on this in the Limitations and Future Directions section (p. 20, Lines 780-788). We acknowledge the relevance of other theoretical perspectives suggested by the reviewer and plan to integrate Self-Determination Theory, Social Exchange Theory, and Algorithmic Management in future research. These frameworks could offer deeper insights into the motivational, social, and organizational aspects of AI dependence and its impact on employee behaviors beyond power dynamics. For instance, Self-Determination Theory could provide a valuable lens to explore how AI's impact on autonomy influences intrinsic motivation and job satisfaction.

For now, we have emphasized how Power-Dependence Theory serves as a solid foundation for our hypotheses and research design, providing a coherent framework to guide our exploration of AI dependence and its effects on leader–employee dynamics. We hope that the refined theoretical positioning is clearer in the revised manuscript (Introduction, pp. 1–4, lines 36-143). We appreciate the reviewer’s helpful suggestion, which has prompted us to reflect on potential avenues for further theoretical expansion in subsequent studies.

Revised Limitations and Future Directions text (p. 20, lines 780-788):

“First, our theoretical framework primarily draws on Power-Dependence Theory (Emerson, 1962) to examine how AI reshapes leader–employee power dynamics. While this theory provides a strong foundation for understanding structural power shifts, it may not fully capture the motivational and social dimensions of AI dependence. Future studies could integrate complementary perspectives—such as Self-Determination Theory (SDT), Social Exchange Theory (SET), or Algorithmic Management frameworks—to offer a more holistic view of how AI affects intrinsic motivation, autonomy, and interpersonal trust. For instance, drawing on SDT may help reveal how AI’s influence on employees’ autonomy supports or hinders their intrinsic motivation and job satisfaction.”

Comments 2: The references are recent and relevant, combining classical theoretical sources with empirical research from 2020–2024. Nonetheless, citation density could be optimized in Section 2.1, where certain classical works are reiterated without adding new interpretive depth.

Response 2: Thank you for your thoughtful feedback on the manuscript. We appreciate your insightful observation regarding the citation density in Section 2.1. We agree that while our references include both classic theoretical sources and more recent empirical studies, some of the classical works are reiterated without adding new interpretive depth.

In response, we have made efforts to refine and optimize the citations in Section 2.1 (p. 4). We have reduced repetition of well-established works and incorporated additional recent studies that expand upon Power Dependence Theory (Emerson, 1962), particularly those that provide fresh perspectives on the application of this framework in the context of AI integration into organizations. This enables us to both maintain a solid theoretical foundation and bring in relevant, up-to-date research that contributes to a more nuanced understanding of power dynamics in modern workplaces.

Furthermore, we have revised the narrative to emphasize the evolving implications of power relationships, especially in light of technological advancements such as AI, which are reshaping traditional leader-employee interactions. By refining our theoretical discussion and streamlining the citation density, we aim to provide a more balanced and concise overview that directly supports the core arguments of the study.

We hope these revisions have addressed your concerns and enhanced the clarity and depth of the section. The updated content is reflected in Section 2.1 of the revised manuscript.

Thank you once again for your constructive comments, and we look forward to any further suggestions you may have.

Revised 2.1. Power dependence theory text (p. 4, lines 146-167):

“In organizational contexts, the relationship between employees and leaders is often shaped by power dynamics, where power is defined as the ability to influence goal attainment and control over valuable resources (Keltner et al., 2003). Traditionally, leaders hold more power due to their higher hierarchical status, which grants them authority over resources such as decision-making, feedback, and task allocation (Krackhardt, 1993; London & Sherman, 2021). Employees, on the other hand, have historically relied on leaders for these resources to accomplish their work-related goals (Wee et al., 2017). According to Power Dependence Theory (Emerson, 1962), power in these relationships is contingent on the dependency between parties—when one party (e.g., the employee) relies on the other (e.g., the leader) for crucial resources, power remains asymmetrical.

However, the introduction of AI into the workplace has shifted these traditional power dynamics(Monod et al., 2023). AI systems, such as intelligent assistants, are increasingly performing functions that were once the exclusive domain of leaders, including decision support, feedback provision, and performance monitoring (Manley & Williams, 2019). This allows employees to access alternative resources, reducing their reliance on leaders and thus altering the traditional power structure. As AI takes on more leadership functions, employees gain greater autonomy and control over their tasks, which di-minishes their dependence on leaders for guidance (Müller & Bostrom, 2016; Walsh, 2018). This shift in power dynamics highlights the need to examine how AI dependence re-shapes leader-employee relationships. By applying Power Dependence Theory, we can better understand how these changes in power dependencies influence employee behaviors, particularly in terms of their interactions with leaders.”

Comments 3: Similarly, the empirical justification for selecting coaching leadership as the moderating variable could be expanded to strengthen its conceptual distinctiveness.

Response 3: Thank you very much for your insightful comments and the suggestion to expand the theoretical justification for selecting coaching leadership as the moderating variable. In the revised manuscript, we have strengthened the theoretical foundation for the selection of coaching leadership and provided a detailed explanation of its unique role in moderating the relationship between AI dependence and employees' sense of power. Below is a detailed outline of how we have further developed the theoretical and empirical basis for this choice:

Firstly, we have enriched the theoretical justification for coaching leadership. According to Power-Dependence Theory (Emerson, 1962), the asymmetry of power in relationships depends on the degree of resource dependency. Leadership behavior serves as a key source of guidance, feedback, and developmental support for employees in achieving their goals. Coaching leadership, which emphasizes employee development and promotes autonomy (Ellinger et al., 2011; Hackman & Wageman, 2005), is highly relevant in the context of AI dependence because it overlaps with the functions provided by AI systems, such as real-time feedback, skill development, and task-specific suggestions. Therefore, the functional overlap between coaching leadership and AI plays a significant role in determining how AI reduces employees' reliance on leaders, diminishing power asymmetry and enhancing employees' sense of power in interactions with their leaders.

Secondly, we have refined the discussion of the functional substitutability between coaching leadership and AI. Coaching leadership is particularly susceptible to substitution by AI, as both share similar functions: providing developmental feedback, task guidance, and skill enhancement. When leaders exhibit high levels of coaching leadership, employees are more likely to view AI as a substitute for leadership functions, such as feedback and guidance. The functional equivalence between AI and coaching leadership further strengthens AI’s role as an alternative resource channel, reducing employees' dependence on human leaders and enhancing their personal sense of power. This dynamic highlights the theoretical uniqueness of coaching leadership as a moderating factor—what matters is not merely the presence of leadership but the degree to which AI can perform equivalent leadership functions. Therefore, when coaching leadership is high, the effect of AI dependence on perceived power is stronger due to the substitutive effect between AI and coaching leadership.

Thirdly, we have incorporated recent empirical studies that support the notion that coaching leadership and AI systems’ capabilities overlap, influencing employees' perceived power. Existing literature (e.g., Brynjolfsson & Mitchell, 2017; Glikson & Woolley, 2020) shows that AI increasingly assumes leadership functions, particularly in providing feedback, supporting decision-making, and facilitating skill development. These studies support the idea that AI can substitute for traditional leadership roles, especially in areas related to skill development and performance feedback.

By integrating these theoretical and empirical foundations, we have further strengthened the theoretical and empirical basis for coaching leadership as a moderating variable. This variable plays a crucial role in influencing the changes in power dynamics between leaders and employees due to AI dependence. We believe these revisions make our theoretical framework more robust and provide deeper insight into the evolving dynamics in leader-employee relationships in the age of AI.

Revised Theory and Hypotheses text (p. 6, lines 253-285):

“According to Power-Dependence Theory (Emerson, 1962), power asymmetry between two parties depends on the control and substitutability of valued resources. When alternative sources become available, dependence on the original resource holder decreases, leading to a redistribution of power. Within organizations, leaders serve as a key source of valued resources—such as developmental guidance, performance feedback, and informational support—that employees rely on to accomplish their goals. Accordingly, when AI begins to replicate or substitute these leadership functions, employees’ dependence on leaders diminishes, enhancing their personal sense of power in interactions with leaders.

Among various leadership styles, coaching leadership is particularly relevant in this dynamic because of its strong focus on employee development and empowerment. Coaching leadership emphasizes supporting employees by establishing open communication channels, providing guidance, and inspiring growth (Eldor & Vigoda-Gadot, 2017; Mäkelä et al., 2024). It offers essential resources and informational support that enable employees to better understand organizational processes and develop greater control over their work environment, thereby boosting engagement and motivation (Chughtai & Buckley, 2011). Through problem-solving assistance, constructive feedback, and soliciting employee input, coaching leaders promote a mutually beneficial exchange that enhances employees’ competence and sustained vigor (Elloy, 2005; Yuan et al., 2019). Moreover, coaching leadership helps employees align personal and organizational goals and clarify role expectations (Borde et al., 2024), creating a developmental climate grounded in trust and autonomy.

However, these very functions—developmental feedback, informational guidance, and empowerment—are also the domains in which AI technologies have become increasingly capable. Modern AI systems can analyze performance data, offer real-time feedback, and generate tailored skill-enhancement suggestions (Brynjolfsson & Mitchell, 2017; Glikson & Woolley, 2020). When leaders demonstrate high levels of coaching leadership, employees perceive greater functional overlap between AI and their supervisors. This functional substitutability amplifies AI’s role as an alternative resource channel, further reducing employees’ dependence on leaders and enhancing their personal sense of power in leader–employee exchanges. Conversely, when coaching leadership is low, the overlap between AI’s capabilities and leadership behaviors is limited, making AI less likely to alter power dynamics.”

We greatly appreciate your helpful suggestions, and these revisions enable us to clarify the unique contribution of coaching leadership as a moderating factor and enhance the rigor of our theoretical framework.

Comments 4: The criteria for participant exclusion and the randomization procedures could be more explicitly described.

Response 4: Thank you for your insightful comments and helpful suggestions. We appreciate your feedback regarding the participant exclusion criteria and the randomization procedures. In response, we have made explicit revisions to clarify these aspects in the revised manuscript.

First, regarding participant exclusion, we have now provided a detailed description of the procedures used to ensure data quality across all studies. As per the previous versions, participants who failed to pass attention checks or submitted incomplete responses were excluded from the analysis. Specifically:

Study 1: We embedded an attention check item and excluded 28 participants who either failed the check or submitted incomplete responses, resulting in a final sample of 172 participants. (p. 8, lines 337-343)

Study 2: Similarly, we excluded 2 participants who failed to pass the attention screening criteria, leading to a final sample of 165 participants. (pp. 9-10, lines 418-423)

Study 3: Participants who did not pass attention checks or submitted incomplete responses were excluded, resulting in a final sample of 218 participants. (p. 11, lines 480-488)

Study 4: Only participants who completed all three waves were included, with a final sample of 285 participants.(pp. 14-15, lines 589-595)

Second, regarding randomization, we have clarified the random assignment procedures used in both Study 1 and Study 2. Specifically:

Study 1: Participants were randomly assigned to either the AI Dependence group (n = 87) or the Control group (n = 85) using Prolific's built-in randomization feature, ensuring unbiased group allocation. (p. 8, lines 350-352)

Study 2: Similarly, participants recruited through the Chinese platform Credamo were randomly assigned to the two conditions using the platform's randomization function, further ensuring balanced group assignment. (p. 10, lines 430-431)

We hope that these revisions provide greater clarity on the criteria for participant exclusion and the randomization procedures. Thank you again for your helpful suggestions, and we look forward to your further feedback.

Comments 5: The discussion could briefly address the potential influence of cultural context (Chinese sample) on both voice expression and power perception, as these constructs are sensitive to cultural dimensions such as collectivism.

Response 5: We sincerely thank you for the insightful comment regarding the influence of cultural context on both voice expression and power perception. We appreciate the suggestion to address the potential role of cultural factors, particularly the cultural differences between collectivism and individualism, in shaping the relationship between AI dependence and employee behavior.

First, in response to your comment, we have expanded the theoretical contributions section to discuss the potential cultural influence on our findings. Although there are cultural differences between China and the United States, we found that our results were consistent across both samples. Specifically, in both cultural contexts, AI dependence positively influenced employees' sense of power and subsequently enhanced voice behavior. This consistency across cultural environments suggests that the impact of AI on power dynamics and voice behavior may transcend cultural boundaries.

We acknowledge that, generally, collectivist cultures (e.g., China) may place a stronger emphasis on group harmony and deference to authority, which could potentially suppress voice behavior (Hui et al., 2004). In contrast, individualistic cultures (e.g., the United States) tend to emphasize autonomy and individual rights, which might encourage more assertive voice behavior (Morrison, 2011). However, our findings indicate that the empowering role of AI could surpass these cultural expectations, suggesting that AI's impact on power perception and voice behavior may have universal effects, regardless of cultural orientation.

Second, in the Limitations and Future Directions section, we have recognized that cultural context likely plays a significant role in shaping how employees from different cultural backgrounds perceive AI's impact on power dynamics. Given that cultural values influence power perceptions and voice behavior, we propose that future research should explore how cultural differences, particularly between individualistic and collectivist cultures, affect AI dependence, power perceptions, and voice behavior. Comparative studies of these cultural contexts will provide a more nuanced understanding of how cultural values shape interactions between employees, leaders, and AI. We believe that examining these cultural factors will offer deeper insights into the global applicability of our findings and contribute to the broader understanding of AI’s role in organizational dynamics across diverse cultural contexts.

Finally, these details have been incorporated into the revised Theoretical Implications section and Limitation and Future direction section (see pp. 18-19 and 20).

Revised Theoretical Implications text (pp. 18-19, lines 718-729):

“Finally, our research also addresses a potential cultural dimension. Despite the differences in cultural contexts between China and the United States, the results in both samples were consistent. In both cultural settings, AI dependence positively influenced employees’ sense of power and subsequently promoted voice behavior. These findings suggest that the impact of AI dependence on power dynamics and behavior may transcend cultural boundaries. This is an intriguing result, as one might expect that collectivist cultures (e.g., China) would show more restrained voice behavior due to the emphasis on group harmony and deference to authority (Hui et al., 2004), whereas individualistic cultures (e.g., the United States) might be more inclined to express voice behavior due to a greater emphasis on autonomy and individual rights (Morrison, 2011).”

Revised Limitation and Future direction text (p. 20, lines 818-826):

“Finally, cultural context likely plays a pivotal role in shaping how employees perceive AI’s influence on power and voice. In high power-distance or collectivist cultures such as China, employees may be less inclined to express voice or challenge authority due to hierarchical norms emphasizing respect and harmony (Hofstede, 2001; Farh et al., 2007; Hui et al., 2004). Future research should systematically compare cultural contexts to explore how national values, institutional systems, and leadership traditions moderate the effects of AI dependence. Such cross-cultural investigations would deepen our understanding of AI’s sociocultural embeddedness and its implications for leadership and employee agency across global settings.”

We hope that these revisions have addressed your concerns and further strengthened our manuscript. Thank you again for your valuable feedback, which has contributed to a more comprehensive exploration of the topic.

Comments 6: Although the study achieves a high level of ecological validity, a more explicit reflection on institutional and hierarchical factors (for example, vertical communication and status orientation in Chinese workplaces) would further contextualize the findings.

Response 6: We sincerely thank you for your valuable feedback. We agree that cultural context, particularly in the Chinese workplace, can have a significant impact on how employees perceive and express power, as well as how they engage in voice behavior. In response to your suggestion, we have expanded the Limitations and Future Directions section to briefly discuss the cultural factors—specifically vertical communication and status orientation—that may influence the relationship between AI dependence, power perception, and voice behavior.

Although our findings were consistent across both the Chinese and U.S. samples, we acknowledge that cultural values such as collectivism in China and individualism in the U.S. may affect how employees perceive the impact of AI on power dynamics and voice behavior. In collectivist cultures, where respect for authority and group harmony are prioritized, employees may be more cautious in voicing opinions or challenging authority. However, our findings suggest that the empowering effects of AI may transcend these cultural norms, potentially making voice behavior more prevalent across different cultures. AI offers employees alternative resources for decision-making and feedback, which may reduce their reliance on leaders and thus encourage more proactive voice behavior, even in more hierarchical cultures.

As such, we believe that future research should explore how cultural differences—particularly comparing individualistic and collectivist cultures—affect AI dependence and its influence on leader-employee relationships and employee behaviors. This would provide valuable insights into how cultural values shape the role of AI in organizational dynamics, further enriching the theoretical understanding of leadership in the AI era.

Lastly, in the revised Limitations and Future Directions section (see page 20, lines 818-826), we propose that future studies examine the role of cultural background in shaping the relationship between AI dependence and employee behaviors, particularly when comparing individualistic and collectivist cultures.

Comments 7: The discussion and practical implications are written with conceptual maturity and intellectual caution. The manuscript avoids technological determinism and presents a balanced, human-centered view of leadership transformation in the AI era. The call for prudence and leader adaptation is well substantiated. Nonetheless, the paper could be further strengthened by including more actionable recommendations, such as guidelines for AI policy design in HRM, leadership development programs, or ethical boundary management in algorithmic decision-making.

Response 7: We truly appreciate the detailed comments and suggestions provided by you. Your constructive feedback has been invaluable in helping us refine and strengthen our work.

We are pleased to hear that the manuscript was considered to present a balanced and human-centered view of leadership transformation in the AI era, and that the intellectual caution and avoidance of technological determinism were noted. In response to the reviewer's suggestions, we have made several revisions to enhance the clarity and actionability of our practical implications. These changes are aimed at providing more concrete and actionable guidelines for organizations and leaders in the AI-augmented workplace.

We have strengthened the Practical Implications section by incorporating more specific recommendations, particularly in the areas of AI policy design, leadership development programs, and ethical boundary management. We now emphasize the importance of developing clear ethical boundaries for AI use, particularly in performance evaluations, recruitment, and feedback processes. AI is presented as a tool to complement, not replace, human leadership, with an emphasis on maintaining human oversight to ensure fairness and transparency. Additionally, we address the need for leadership development programs that help leaders integrate AI into their practices, focusing on human-centered skills like empathy, moral judgment, and trust-building, which AI cannot replicate.

We also further highlighted the importance of leadership adaptation in the AI era. While AI enhances employees' sense of power and promotes voice behavior, we recognize that it can also shift power dynamics within organizations. To manage this shift effectively, leaders must adapt by focusing more on fostering human connection and ethical judgment rather than just task control. We now emphasize that AI should serve as a partner to human leadership, with leaders playing a key role in maintaining relational dynamics and ensuring AI’s ethical and inclusive application.

These revisions aim to ensure that the practical implications are not only conceptually rigorous but also actionable, offering clear guidance for organizations to integrate AI effectively while preserving essential human leadership qualities (see page 19, lines 731–776).

Revised Practical Implications text (p. 19, lines 731-776):

“Our study demonstrates that AI dependence can enhance employees’ sense of power, which in turn promotes voice behavior in the workplace. However, this empowerment dynamic also creates challenges for organizational leadership. While AI can strengthen employees’ confidence and initiative, it may simultaneously weaken the perceived authority and influence of leaders if not managed effectively. These findings provide several important implications for how organizations and leaders can adapt to the realities of the AI-augmented workplace.

First, even in the era of advanced AI applications, leadership remains indispensable (Agrawal et al., 2017; Davenport & Kirby, 2016; Kolbjørnsrud et al., 2017; Wilson & Daugherty, 2018). As AI increasingly undertakes decision-support and feedback functions, leaders must adapt by developing capabilities that complement, rather than compete with, intelligent systems. Our findings suggest that when leaders display strong coaching leadership behaviors, their functions often overlap with AI’s analytical and feedback capabilities. This overlap calls for leaders to redefine their roles—placing less emphasis on task control and more on fostering human connection, interpretation, and ethical judgment. Leading responsibly in AI-augmented contexts requires maintaining human oversight and accountability across all stages of AI application (Hossain et al., 2025). By improving AI literacy and ethical awareness, leaders can ensure fairness, transparency, and auditability in algorithmic decision-making, guarding against biases and promoting inclusive and equitable outcomes.

Second, organizations should establish clear ethical boundaries and operational guidelines to govern the use of AI, particularly in areas such as performance evaluation, recruitment, and feedback delivery. AI should be treated as a supportive tool that enhances—rather than replaces—human leadership. Maintaining human oversight in AI-supported decision processes is essential to safeguard fairness and transparency. In addition, organizations should regularly assess how AI influences employees’ autonomy, creativity, and engagement to prevent overreliance on algorithmic systems from undermining intrinsic motivation. Ethical frameworks should also protect employees’ opportunities for voice and participation, ensuring that AI empowers rather than silences them.

Third, leadership development programs should evolve to reflect the new skill sets required in AI-enabled organizations. Leaders need to learn how to integrate AI technologies into their management practices in ways that strengthen both employee empowerment and relational trust. Training should emphasize the uniquely human aspects of leadership—empathy, moral judgment, and meaning-making—that AI cannot replicate. By doing so, leaders can maintain their relevance and authority while leveraging AI to improve efficiency, insight, and responsiveness.

Finally, organizations should adopt a balanced approach to AI integration—one that recognizes both its empowering potential and its relational risks (Newman et al., 2020). Our findings highlight that AI dependence may amplify employees’ perceived power, especially under strong coaching leadership, which can subtly shift traditional power dynamics in the workplace. To manage this shift productively, leaders must focus on building collaborative relationships and shared accountability, positioning AI as a catalyst for collective growth rather than as a substitute for leadership. Through deliberate policy design, continuous ethical review, and leadership adaptation, organizations can ensure that AI and human leadership work synergistically to foster both employee voice and organizational vitality in the digital era.”

Once again, we would like to thank you and the reviewers for your invaluable feedback. We hope the revised manuscript addresses all concerns and meets the expectations of the reviewers. We look forward to your continued guidance in the review process.

Comments 8: Overall, this is a well-conceived, empirically grounded, and theoretically original contribution. With modest revisions aimed at deepening theoretical integration and refining cultural contextualization, the paper would be well positioned for publication in a leading journal like Behavioral Sciences. I commend the authors for their rigorous work and wish them continued success in their research endeavors.

Response 8: We sincerely appreciate your positive overall assessment of our manuscript and your recognition of its conceptual rigor, empirical grounding, and theoretical originality. Your encouraging comments are deeply motivating. At the same time, we fully acknowledge your constructive suggestions regarding the need to further deepen the theoretical integration and strengthen the cultural contextualization. In response, we have carefully revised the relevant sections to clarify the theoretical mechanisms, enrich the cross-cultural discussion, and more explicitly position our contribution within the broader literature. Thank you again for your thoughtful and generous feedback. Your guidance has greatly improved the quality of the manuscript, and we remain committed to further refining the work to meet the standards of Behavioral Sciences.

Reviewer 4 Report

Comments and Suggestions for Authors

This paper examines the effect of AI dependence on voice behavior. It captures a timely emerging phenomenon, is grounded in solid theoretical logic, and provides diverse empirical evidence. I believe it is a strong and meaningful study. However, I recommend addressing the following theoretical concern in both the main text and the discussion section.

While I agree with the authors’ argument that greater AI dependence may increase individuals’ sense of power, it seems equally plausible that such heightened power could lead individuals to act independently rather than engage in voice behavior. In a similar vein, increased AI dependence may reduce interactions and exchanges with supervisors, which could in turn diminish the likelihood of voice behavior. This should be discussed more explicitly. Importantly, this is not merely a theoretical consideration. If there are relevant variables in your dataset, the authors should conduct empirical analyses to examine these possibilities.

Author Response

Comments 1: This paper examines the effect of AI dependence on voice behavior. It captures a timely emerging phenomenon, is grounded in solid theoretical logic, and provides diverse empirical evidence. I believe it is a strong and meaningful study. However, I recommend addressing the following theoretical concern in both the main text and the discussion section.

While I agree with the authors’ argument that greater AI dependence may increase individuals’ sense of power, it seems equally plausible that such heightened power could lead individuals to act independently rather than engage in voice behavior. In a similar vein, increased AI dependence may reduce interactions and exchanges with supervisors, which could in turn diminish the likelihood of voice behavior. This should be discussed more explicitly. Importantly, this is not merely a theoretical consideration. If there are relevant variables in your dataset, the authors should conduct empirical analyses to examine these possibilities.

Response 1: We are grateful for your thoughtful comments regarding the potential alternative mechanisms linking AI dependence to voice behavior. Specifically, we recognize the concern that increased AI dependence could lead to greater independence in employees’ behavior, potentially diminishing the likelihood of voice behavior. Additionally, the suggestion that AI dependence might reduce interactions with supervisors, thereby diminishing voice behavior, is an important consideration.

In response, we have explicitly addressed this possibility in the Limitations and Future Directions section of the manuscript. While our study primarily focuses on how AI dependence enhances employees’ sense of power and promotes voice behavior, we acknowledge that AI dependence could also lead to more independent behavior or reduce interactions with supervisors, which may subsequently weaken voice behavior.

To address this, we suggest that future research could explore alternative pathways by incorporating variables related to leader–employee interaction frequency, feedback reliance, and employee autonomy. This would help examine whether AI dependence reduces voice behavior by weakening interpersonal dependence on leaders. We believe that these considerations open up new directions for research and will be important for understanding the nuanced effects of AI on employee behaviors.

Regarding the empirical analysis of these potential pathways, although our current dataset does not include specific variables related to leader–employee interaction frequency or autonomy, we recognize the importance of investigating these relationships further. We have added a suggestion in the 12.3. Limitations and Future Directions section (see page 20, lines 799-812) to explore these alternative mechanisms in future studies.

We hope the revisions we made adequately address your concerns and strengthen the manuscript. By incorporating a more nuanced discussion of how AI dependence may lead to reduced interactions with supervisors and more independent behavior, we have clarified the potential complexity of the relationship between AI, power dynamics, and voice behavior. We also appreciate the opportunity to suggest future directions for research that will help explore these important issues further.

Revised Limitations and Future Directions text (p. 20, lines 789-802):

“Second, although this study identifies employees’ sense of power as the central psychological mechanism linking AI dependence to voice behavior, alternative pathways may also explain this relationship. AI dependence could alter affective or motivational states such as confidence, anxiety, or fatigue, which in turn influence employees’ willingness to speak up. Future research may examine indicators of leader–employee interaction (e.g., communication frequency, feedback reliance) to determine whether heightened AI dependence reduces upward voice by weakening interpersonal reliance on leaders. Furthermore, while this study focuses on voice behavior as a key relational outcome, AI dependence may also shape other behaviors—such as creativity, risk-taking, or opportunism (Boussioux et al., 2024; Jia et al., 2024). Greater autonomy and information access might foster innovation, whereas excessive reliance on AI could generate isolation, alienation, or self-serving tendencies (Hai et al., 2025). Exploring these dual consequences would enrich understanding of how AI dependence reshapes both social and task-oriented behaviors in organizations.”

Once again, thank you for your valuable feedback. We look forward to your continued guidance as we move forward with the review process.

Round 2

Reviewer 2 Report

Comments and Suggestions for Authors

The comments have been addressed up to certain level. The English could be improved to more clearly express the research.

Author Response

Comment1:

The comments have been addressed up to certain level. The English could be improved to more clearly express the research.

Response1:

Thank you for your time in reviewing our revised manuscript and for your valuable feedback.

We fully agree with your comment regarding the need for further improvement in the English language to enhance clarity. To address this point thoroughly, we have enlisted the services of a professional English language editing company to perform a comprehensive language polish on the entire manuscript.

The native English-speaking editors, who have expertise in our field, have refined the manuscript for grammar, sentence structure, word choice, and overall clarity of academic expression. We are confident that this professional editing has significantly improved the readability and quality of the manuscript, allowing the research to be expressed more clearly.

For your reference, we have attached the certificate of editing provided by the service.

We thank you again for your comment, which has helped us improve our work. We hope the language in the current version now meets the required standard.
